# ReMatching Dynamic Reconstruction Flow

**Sara Oblak** [1] **Despoina Paschalidou** [1] **Sanja Fidler**[1 2 3] **Matan Atzmon** [1]
[1] NVIDIA   [2] University of Toronto   [3] Vector Institute
{soblak,dpaschalidou,sfidler,matzmon}@nvidia.com

## Abstract

Reconstructing a dynamic scene from image inputs is a fundamental computer vision task with many downstream applications. Despite recent advancements, existing approaches still struggle to achieve high-quality reconstructions from unseen viewpoints and timestamps. This work introduces the ReMatching framework, designed to improve reconstruction quality by incorporating deformation priors into dynamic reconstruction models. Our approach advocates for velocity-field-based priors, for which we suggest a matching procedure that can seamlessly supplement existing dynamic reconstruction pipelines. The framework is highly adaptable and can be applied to various dynamic representations. Moreover, it supports integrating multiple types of model priors and enables combining simpler ones to create more complex classes. Our evaluations on popular benchmarks involving both synthetic and real-world dynamic scenes demonstrate that augmenting current state-of-the-art methods with our approach leads to a clear improvement in reconstruction accuracy.

## 1 Introduction

This work addresses the challenging task of novel-view dynamic reconstruction. That is, given a set of images of a dynamic scene evolving over time, the task objective is to render images from any novel view or intermediate point in time. Despite significant progress in recent years (Lombardi et al., 2021; Fridovich-Keil et al., 2023; Yunus et al., 2024), effectively learning representations of dynamic scenes still remains an open challenge. The main hurdle arises from the typically sparse nature of multi-view inputs, both temporally and spatially. To address this, prior knowledge is often incorporated - either from a physical prior such as rigidity (Sorkine & Alexa, 2007), or data-driven priors derived from large foundation models (Ling et al., 2024; Wang et al., 2024). However, existing approaches struggle with an inherent trade-off: while priors improve generalization, they often compromise reconstruction fidelity by failing to match exactly the given input. In turn, designing a method that effectively integrates priors without sacrificing high-fidelity reconstructions remains unresolved.

To address this issue, we introduce the ReMatching framework, a novel approach for designing and integrating deformation priors into dynamic reconstruction models. The ReMatching framework has three core features: i) optimization objective that aligns reconstruction solutions with prior regularization as closely as possible, without sacrificing fidelity; ii) ensured applicability to various model functions, including time-dependent rendered pixels or particles representing scene geometry; and, iii) provide a flexible design of deformation prior classes, allowing more complex classes to be built from simpler ones.

To support the usage of rich deformation prior classes, we advocate for priors expressed through velocity fields. A velocity field is a mathematical object that describes the instantaneous change in time the deformation induces. As such, a velocity field can potentially provide a simpler characterization of the underlying *flow* deformation. For example, the complex class of volume-preserving flow deformations is characterized by the condition of being generated by divergence-free velocity fields (Eisenberger et al., 2019). However, representing a deformation through its generating velocity field typically necessitates numerical simulation for integration, a procedure that can be computationally expensive and time-consuming. Nevertheless, recent progress in flow-based generative models (Ben-Hamu et al., 2022; Lipman et al., 2022; Albergo et al., 2023) supports simulation-free flow training, inspiring this work to explore simulation-free training for flow-based dynamic reconstruction models. Therefore, our framework is specifically designed to integrate with dynamic reconstruction models that represent dynamic scenes directly through time-dependent reconstruction functions (Pumarola et al., 2021; Yang et al., 2023).

Exploiting the simplicity offered by velocity-field-based deformation prior classes, we observe that the *projection* of a time-dependent reconstruction function onto a velocity-field prior class can be framed as a flow-matching problem. The opportunity to access the projected flow is reminiscent of the Alternating Projections Method (APM) (Deutsch, 1992), a greedy algorithm *guaranteed* in finding the closest points between two sets. Therefore, we suggest an optimization objective aimed at re-projecting back onto the set of reconstruction flows. This corresponds to a flow-matching loss that we term the *ReMatching* loss. Our hypothesis is that by mimicking the APM, this optimization would converge to solutions that not only meet the reconstruction objective, but also reach the *closest* possible alignment to the required prior class. By doing so, we achieve the desired goal of improving generalization to unseen timestamps without compromising solutions' fidelity levels.

We instantiate our framework with a dynamic model based on the popular Gaussian Splats (Kerbl et al., 2023) rendering model. We explore several constructions for deformation prior classes including piece-wise rigid and volume-preserving deformations. Additionally, we demonstrate our framework's usability for two different types of time-dependent functions: rendered image pixels color, and particle positions representing scene geometry. Lastly, we evaluate our framework on standard dynamic reconstruction benchmarks, involving both synthetic and real-world scenes, and showcase clear improvement in reconstruction quality.

**Our contributions.**  In summary, the main contributions of this paper are:

1. We propose the ReMatching framework, which controls the optimization of dynamic reconstruction models to converge to solutions that closely align with a predefined prior class of deformations. In turn, the framework balances achieving high-fidelity reconstructions with leveraging the benefits of adhering to prior assumptions.

2. The framework unifies various types of model functions, including geometry representations and image rendering, under a single cohesive approach, ensuring broad applicability and making future advancements within it relevant to a wide range of models.

3. The framework allows for the combination of multiple prior classes, enabling users to design and adapt the method for their specific reconstruction settings.

## 2 RELATED WORK

**Flow-based 3D dynamics.**  There is an extensive body of works utilizing flow-based deformations for 3D related problems. For shape interpolation, (Eisenberger et al., 2019) considers volume-preserving flows. For dynamic geometry reconstruction, (Niemeyer et al., 2019) suggests learning neural parametrizations of velocity fields. This representation is further improved by augmenting it with a canonicalized object space parameterization (Rempe et al., 2020; Ren et al., 2021) or by simultaneously optimizing for 3D reconstruction and motion flow estimation (Vu et al., 2022). Similarly to (Niemeyer et al., 2019), (Du et al., 2021) suggests flow-based representation of dynamic rendering model based on a neural radiance field (Mildenhall et al., 2020). More recently, (Chu et al., 2022; Yu et al., 2023) explores combining a time-aware neural radiance field with a velocity field for modelling fluid dynamics. In contrast to our framework they focus exclusively on recovering the deformation of specific fluids i.e. smoke and not on reconstructing generic non-rigid objects.

**Dynamic novel-view rendering models.**  Neural Radiance Fields (NeRF) (Mildenhall et al., 2020) is a popular image rendering model combining an implicit neural network with volumetric rendering. Several follow-up works (Pumarola et al., 2021; Park et al., 2021a; Tretschk et al., 2021) explore using NeRF for non-rigid reconstruction, by optimizing for time-dependent deformations. More recently, several works (Fridovich-Keil et al., 2023; Cao & Johnson, 2023; Wu et al., 2023; Song et al., 2023; Guo et al., 2023) try to address the training and inference inefficiencies of continuous volumetric representations by incorporating planes and grids into a spatio-temporal NeRF. An alternative to NeRF, suggesting an explicit scene representation, is the Gaussian Splatting (Kerbl et al., 2023) rendering model. Several works incorporate dynamics with Gaussian Splatting. (Yang et al., 2023) introduce a time-conditioned local deformation network. Similarly, (Wu et al., 2023) also relies on a canonical representation of a scene but further improves efficiency by considering a deformation model based on on $k$-planes (Fridovich-Keil et al., 2023). Lastly, (Lu et al., 2024) propose the integration of a global deformation model.

## 3 METHOD

Given a collection of images, $F_t = \left\{ I_i^t \right\}_{i=1}^M$, captured at $T$ time steps, from $M \geq 1$ viewing directions, we seek to develop an image-based model for novel-view synthesis that can effectively render new images from unseen viewpoints in any direction $\boldsymbol{d} \in \mathcal{S}^2$ and any time $t \in [t_1, t_T]$. Since we aim to support several time-dependent elements in a dynamic reconstruction model, we employ a general notation for a dynamic image model. That is,

$$t \mapsto \Psi_t = \left\{ \psi(t) \,|\, \psi : \mathbb{R}_+ \to V \right\}, \tag{1}$$

with $\Psi_t$ representing the evaluation at time $t$ of all of the model components. Each element function $\psi : \mathbb{R}_+ \to V$, where $V$ is a vector space, can specify any time-dependent quantity specified by the model. $V$ denotes a different vector space depending on what $\psi$ models. For instance, if $\psi$ models time-dependent image pixels RGB color, $V = C^1(\mathbb{R}^d) = \left\{ f | f : \mathbb{R}^d \to \mathbb{R}^3, \nabla f \text{ exists and continuous} \right\}$ with $d = 2$. Whereas, if $\psi$ models the time-dependent position of $n$ particles representing the underlying scene geometry, $V = \mathbb{R}^{n \times d}$ with $d = 3$. Lastly, in what follows, we interchangeably switch between the notations $\psi(t)$ and $\psi_t$.

The common scheme to learning $\Psi_t$ involves supervising the model's image predictions at the given timestamps to reconstruct the input images $F_t$. The specific details of the time-dependent reconstruction model $\Psi_t$ and the reconstruction loss are deferred to Section 5. We begin first by introducing our proposed framework for incorporating priors through velocity fields.

### 3.1 VELOCITY FIELDS

We consider a velocity field to be a time-dependent function of the form:

$$v : \mathbb{R}^d \times \mathbb{R}_+ \to \mathbb{R}^d, \tag{2}$$

where usually $d = 3$ or $d = 2$. A velocity field defines a time-dependent deformation in space $\phi_t : \mathbb{R}^d \to \mathbb{R}^d$, also known as a *flow*, via an Ordinary Differential Equation (ODE):

$$\begin{cases} \dfrac{\partial}{\partial t} \phi_t(\boldsymbol{x}) = v(\phi_t(\boldsymbol{x}), t) \\ \phi_0(\boldsymbol{x}) = \boldsymbol{x}. \end{cases} \tag{3}$$

Flow-based deformations are an ubiquitous modeling tool (Rezende & Mohamed, 2015; Chen et al., 2018) that has been extensively used in various dynamic reconstruction tasks (Niemeyer et al., 2019; Du et al., 2021). In a dynamic reconstruction model, a flow deformation can be incorporated by defining a time-dependent function $\psi_t : \mathbb{R}^d \to \mathbb{R}$ as a push-forward of some reference function $\psi_0$, i.e., $\psi_t = \phi_{t*}\psi_0$. A key advantage of a flow-based deformation model is that its generating velocity field often admits simple characterizations, facilitating the integration of priors into the model. For example, restricting $\phi_t$ to be volume-preserving can be achieved by imposing the constraint $\mathrm{div}(v) = 0$ (Eisenberger et al., 2019).

However, recovering $\psi_t$ values in the case $\psi_t = \phi_{t*}\psi_0$ is not explicit. Typically, this is achieved by solving the continuity equation [1]

$$\frac{\partial}{\partial t} \psi_t(\boldsymbol{x}) + \mathrm{div} \left( \psi_t(\boldsymbol{x}) v_t(\boldsymbol{x}) \right) = 0, \, \forall \boldsymbol{x} \in \mathbb{R}^d, \tag{4}$$

which necessitates a numerical simulation. This introduces computational challenges for training flow-based models, as errors in the numerical simulation can destabilize the optimization process. Therefore, to overcome this hurdle, our framework assumes a reconstruction model consisting of functions $\psi_t$ that are simulation-free, i.e., each evaluation of $\psi_t$ requires only a single step. However, since we advocate for a deformation prior class formulation based on velocity fields, a key challenge lies in controlling a simulation-free $\psi_t$ to adhere to such a prior. Addressing this challenge is a central aspect of our framework, as outlined in the following section.

---

[1]Assuming $\psi_t$ obeys a conservation law, where $v$ continuously deforms $\psi_t$.

## 3.2 FLOW REMATCHING

We assume that for a time-dependent reconstruction function, $\psi_t \in \Psi_t$, there exists an underlying flow $\phi_t$ such that $\psi_t$ can be described as a push-forward by $\phi_t$. We refer to $\phi_t$ as the *reconstruction flow* and denote its generating velocity field by $v_t$. Under our assumption that $\psi_t$ is simulation-free, neither $\phi_t$ nor $v_t$ is directly accessible. Nevertheless, for now, we assume access to an element $v_t \in \mathcal{V}$, where $\mathcal{V}$ represents the set of all the possible reconstruction-generating velocity fields. This assumption will be relaxed later. Let $\mathcal{P} \subset \left\{ u_t \mid u : \mathbb{R}^d \times \mathbb{R}_+ \to \mathbb{R}^d \right\}$ be a prior class of velocity fields to which $v_t$ should belong. In Section 4, we discuss various choices for the prior class $\mathcal{P}$.

In some of the choices for $\mathcal{P}$, requiring $v_t \in \mathcal{P}$ could be over-restrictive, conflicting with the fact that $v_t$ also adheres to generate the reconstruction flow. Hence, an appealing objective would be to optimize $v_t$ so that it is the closest element to $\mathcal{P}$ out of the set $\mathcal{V}$. We suggest an optimization procedure mimicking the alternating projections method (APM) (Deutsch, 1992). The APM is an iterative procedure where alternating orthogonal projections are performed between two closed Hilbert sub-spaces $V$ and $P$. Specifically, $v^{k+1} = \mathrm{proj}_V \left( \mathrm{proj}_P(v^k) \right)$ guarantees the convergence of $v^k$ to $\mathrm{dist}(V, P)$. Following this concept, our next step is to find a suitable notion for defining the projection operator for reconstruction generating velocity fields.

Since $v_t$ is unknown in our case, we suggest leveraging the continuity equation (4), which provides both a sufficient and a *necessary* condition for the generating velocity field of $\phi_t$ in terms of $\psi_t$ and its partial derivatives. Specifically, we introduce a projection procedure to recover the closest element in the prior class to the reconstruction flow by solving the following matching optimization problem:

$$u(\cdot, t) = \arg\min_{u_t \in \mathcal{P}} \rho(u_t, \psi_t), \tag{5}$$

where,

$$\rho(u_t, \psi_t) = \int \left| \frac{\partial}{\partial t} \psi_t(\boldsymbol{x}) + \mathrm{div}\left( \psi_t(\boldsymbol{x}) u_t(\boldsymbol{x}) \right) \right|^2 \, d\boldsymbol{x}. \tag{6}$$

This procedure is illustrated in the right inset, where $u_t$ (red dot) is the closest point to $v_t$ on $\mathcal{P}$. Notably, neither $\phi_t$ nor $v_t$ appear in equation 5, aligning with our objective of relying solely on $\psi_t$.

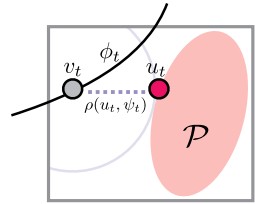

Following the alternating projections concept, the matched $u_t$ should be projected back onto $\mathcal{V}$ to propose a better candidate for $v$. This corresponds to a flow matching problem in $u_t$. We refer to this procedure as ReMatching and introduce the flow ReMatching loss, $L_{\mathrm{RM}}$, a matching loss striving for the reconstruction flow to match $u_t$. That is,

$$L_{\mathrm{RM}}(\boldsymbol{\theta}) = \mathbb{E}_{t \sim U[0,1]} \rho(u_t, \psi_t) \tag{7}$$

where $u_t$ is a solution of equation 5 and $\boldsymbol{\theta}$ denotes *solely* the parameters of $\psi_t$.

---

**Algorithm 1** ReMatching loss

---

**Require:** Solver for 5, times $\{t_l\}$
$L_{\mathrm{RM}} = 0$
**for** $t \in \{t_l\}$ **do**
    $u_t(\cdot) \leftarrow \texttt{solve}(\rho, \psi_t(\cdot))$
    $L_{\mathrm{RM}} \leftarrow L_{\mathrm{RM}} + \rho(u_t(\cdot), \psi)$
**end for**
**Return:** $L_{\mathrm{RM}}$

---

The ReMatching loss is designed to supplement a reconstruction loss $L_{\mathrm{REC}}$ on the model parameters $\boldsymbol{\theta}$. Thus, our framework's final loss for dynamic reconstruction training is

$$L(\boldsymbol{\theta}) = L_{\mathrm{REC}}(\boldsymbol{\theta}) + \lambda L_{\mathrm{RM}}(\boldsymbol{\theta}) \tag{8}$$

where $\lambda > 0$ is a hyper-parameter. Note that in practice, the integral in equation 7 is approximated by a sum using random samples $\{t_l\} \sim U[0,1]$. In addition, for the ReMatching procedure to be seamlessly incorporated into a reconstruction training process, it is essential that problem 5 can be solved efficiently. To support this, in section 5 we provide linear constructions for $\mathcal{P}$ that are sufficiently expressive to encompass the considered prior classes while allowing efficent solutions to equation 5. Algorithm 1 summarises the details of computing 7. Note that calculating $\nabla_{\boldsymbol{\theta}} L_{\mathrm{RM}}$ does *not* necessarily require the cumbersome calculation of $\nabla_{\boldsymbol{\theta}} \arg\min_{u_t \in \mathcal{P}} \rho(u(\cdot, t), \psi_t)$, since according to Danskin's theorem (Madry, 2017), $\nabla_{\boldsymbol{\theta}} \rho(u_t, \psi_t) = \nabla_{\boldsymbol{\theta}} \min_{u_t \in \mathcal{P}} \rho(u_t, \psi_t)$. Additional implementation details regarding the losses can be found in the Appendix.

## 4 FRAMEWORK INSTANCES

This section presents several instances of the ReMatching framework discussed in this work. One notable setting is when $V = \mathbb{R}^{n \times d}$, i.e., $\psi_t = (\gamma_t^1, \cdots, \gamma_t^n)^T$, where each $\gamma^i : \mathbb{R}_+ \to \mathbb{R}^d$. In this case, equation 6 becomes:

$$\rho(u_t, \psi_t) = \sum_{i=1}^{n} \left\| u_t(\gamma_t^i) - \frac{d}{dt} \gamma_t^i \right\|^2. \tag{9}$$

Details of this derivation can be found in section 8.1.2. For the settings where $V = C^1(\mathbb{R}^d)$, equation 6 involves the computation of a spatial integral, which can be approximated by sampling a set of points $\{\boldsymbol{x}_i\}_{i=1}^n$. Moreover, taking into account that all prior classes incorporated in this work are divergence-free, equation 6 becomes:

$$\rho(u_t, \psi_t) = \sum_{i=1}^{n} \left| \frac{\partial}{\partial t} \psi_t(\boldsymbol{x}_i) + \langle \nabla \psi_t(\boldsymbol{x}_i), u_t(\boldsymbol{x}_i) \rangle \right|^2, \tag{10}$$

since $\operatorname{div}(\psi_t(\boldsymbol{x}) u_t(\boldsymbol{x})) = \langle \nabla_x \psi_t(\boldsymbol{x}), u_t(\boldsymbol{x}) \rangle + \psi_t(\boldsymbol{x}) \operatorname{div} u_t(\boldsymbol{x})$.

We now formulate several useful prior classes of velocity fields $\mathcal{P}$. A key feature of all the following constructions is their reliance on linear parameterizations, capitalizing on the fact that linear subspaces are sufficiently expressive to represent the velocity-based prior classes considered. This approach enables the use of efficient solvers for problem 5, reducing the computational task to solving a system of $d$ linear equations, with a run-time complexity of at most $O(n)$.

### 4.1 PRIOR DESIGN

**Directional restricted deformation.** In certain scenarios, it is safe to assume that the reconstruction flow can only deform along specific directions. For example, in an indoor scene, where furniture is placed on the floor, deformations would typically occur only in directions parallel to the floor plane. Let $\boldsymbol{v} \in \operatorname{span}\{\boldsymbol{v}_1, \cdots, \boldsymbol{v}_l\}$, $1 \le l \le d$ and $\{\boldsymbol{v}_1, \cdots, \boldsymbol{v}_l\}$ is a predefined orthonormal basis in which the flow remains static. Then, the prior class becomes:

$$\mathcal{P}_I = \{u_t | \langle u(\boldsymbol{x}, t), \boldsymbol{v}_m \rangle = 0, \forall m \in [l]\}. \tag{11}$$

When considering the matching minimization problem 5 in the settings of equation 9, we get:

$$\min_{u_t \in \mathcal{P}_I} \sum_{i=1}^{n} \left\| u_t(\gamma_t^i) - \frac{d}{dt} \gamma_t^i \right\|^2 = \sum_{i=1}^{n} \left\| \boldsymbol{V}^T \frac{d}{dt} \gamma_t^i \right\|^2, \tag{12}$$

where $\boldsymbol{V} = [\boldsymbol{v}_1, \cdots \boldsymbol{v}_l]$. For the settings involving equation 10, the matching minimization problem is solved by:

$$\min_{u_t \in \mathcal{P}_I} \sum_{i=1}^{n} \left| \frac{\partial}{\partial t} \psi_t(\boldsymbol{x}_i) + \langle \nabla \psi_t(\boldsymbol{x}_i), u_t(\boldsymbol{x}_i) \rangle \right|^2 = \sum_{i=1}^{n} \frac{\partial}{\partial t} \psi_t(\boldsymbol{x}_i)^2 \left( 1 - \frac{\langle \nabla \psi_t(\boldsymbol{x}_i), \boldsymbol{V}_* \nabla \psi_t(\boldsymbol{x}_i) \rangle}{\|\nabla \psi_t(\boldsymbol{x}_i)\|^2} \right)^2, \tag{13}$$

where $\boldsymbol{V}_* = (I - \boldsymbol{V}\boldsymbol{V}^T)$.

**Rigid deformation.** One widely used prior in the dynamic reconstruction literature is rigidity, i.e., objects in a scene can only be deformed by a rigid transformation consisting of a translation and an orthogonal transformation. In a simple case, where it is assumed that the underlying dynamics consists of *one* rigid motion, the reconstruction flow would be of the form

$$\gamma(t) = \boldsymbol{R}(t)\boldsymbol{x}_0 + \boldsymbol{b}(t) \tag{14}$$

with $\boldsymbol{R}(t) \in O(3)$ and $\boldsymbol{b}(t) \in \mathbb{R}^3$. Differentiating $\gamma$ and solving for $\boldsymbol{x}_0$ yields that

$$\frac{d}{dt}\gamma(t) = \dot{\boldsymbol{R}}(t)\boldsymbol{R}^T(t)(\gamma(t) - \boldsymbol{b}(t)) + \dot{\boldsymbol{b}}(t). \tag{15}$$

Since $\dot{\boldsymbol{R}}(t)\boldsymbol{R}^T(t)$ is a skew-symmetric matrix, we suggest the following natural parameterization for the prior class

$$\mathcal{P}_{II} = \{u_t | u(\boldsymbol{x}, t) = \boldsymbol{A}_t \boldsymbol{x} + \boldsymbol{b}_t, \boldsymbol{A}_t \in \mathbb{R}^{d \times d}, \boldsymbol{A}_t = -\boldsymbol{A}_t^T, \boldsymbol{b}_t \in \mathbb{R}^d\}. \tag{16}$$

Substituting $\mathcal{P}_{II}$ in problem 5 using equation 9 yields the following minimization problem:

$$\min_{(\boldsymbol{A}_t, \boldsymbol{b}_t)} \sum_{i=1}^{n} \left\| \boldsymbol{A}_t \gamma_t^i + \boldsymbol{b}_t - \frac{d}{dt} \gamma_t^i \right\|^2 \quad \text{s.t. } \boldsymbol{A}_t = -\boldsymbol{A}_t^T. \tag{17}$$

For the settings involving equation 10, the minimization problem 5 becomes:

$$\min_{(\boldsymbol{A}_t, \boldsymbol{b}_t)} \sum_{i=1}^{n} \left| \frac{\partial}{\partial t} \psi_t(\boldsymbol{x}_i) + \langle \nabla \psi_t(\boldsymbol{x}_i), \boldsymbol{A}_t \boldsymbol{x}_i + \boldsymbol{b}_t \rangle \right|^2 \quad \text{s.t. } \boldsymbol{A}_t = -\boldsymbol{A}_t^T. \tag{18}$$

Importantly, both 17 and 18 are constrained least-squares problems. Thus, as detailed in Lemma 1, they enjoy an analytic solution that can be computed efficiently.

**Volume-preserving deformation.** So far we have only covered prior classes that may be too simplistic for capturing complex real-world dynamics. To address this, a reasonable assumption would be to include deformations that preserve the volume of any subset of the space. Notably, the rigid deformations prior class discussed earlier strictly falls within this class as well. Interestingly, volume-preserving flows are characterized by being generated via a divergence-free velocity field, i.e., $\text{div } u = 0$. To this end, we propose the following prior class:

$$\mathcal{P}_{III} = \left\{ u_t \mid u_t(\boldsymbol{x}) = \sum_{j=1}^{k} \beta_j b_j(\boldsymbol{x}), \boldsymbol{\beta} = [\beta_1, \cdots, \beta_k]^T \in \mathbb{R}^k \right\}, \tag{19}$$

where for each basis $b_j : \mathbb{R}^d \to \mathbb{R}^d$, we assume that $\text{div}(b_j) = 0$. Clearly, $\text{div}(u_t) = 0$ for any choice of $\boldsymbol{\beta} \in \mathbb{R}^k$. Taking into account that $\text{div curl } u = 0$, we follow Eisenberger et al. (2019), and incorporate the following basis functions:

$$b_j(\boldsymbol{x}) \in \left\{ \text{curl}\left( \phi_j(\boldsymbol{x}) e_1^T \right), \cdots, \text{curl}\left( \phi_j(\boldsymbol{x}) e_d^T \right) \right\}, \tag{20}$$

where $\phi_j : \mathbb{R}^d \to \mathbb{R}$, $\phi_j(\boldsymbol{x}) = \prod_{l=1}^{d} \sin\left( j_l \pi e_l^T \boldsymbol{x} \right)$ with $j_l \in \mathbb{N}$ denoting the frequency for the $l^{\text{th}}$ coordinate of the $j^{\text{th}}$ basis function. Combining this prior with equation 9, yields the following minimization problem:

$$\min_{\boldsymbol{\beta}} \sum_{i=1}^{n} \left\| \sum_{j=1}^{k} \beta_j b_j(\gamma_t^i) - \frac{d}{dt} \gamma_t^i \right\|^2. \tag{21}$$

Similarly, for the case of equation 10, we get:

$$\min_{\boldsymbol{\beta}} \sum_{i=1}^{n} \left| \frac{\partial}{\partial t} \psi_t(\boldsymbol{x}_i) + \left\langle \nabla \psi_t(\boldsymbol{x}_i), \sum_{j=1}^{k} \beta_j b_j(\boldsymbol{x}_i) \right\rangle \right|^2. \tag{22}$$

In particular, both minimization problems of 21 and 22 correspond to a standard least-squares problem and have an analytic solution that can be efficiently computed. A key decision involved in using the prior class $\mathcal{P}_{III}$ is to select the number of basis functions $k$. However, setting $k$ equal to a large value would make $\mathcal{P}_{III}$ overly permissive, effectively neutralizing the ReMatching loss. To address this, in what follows, we propose an additional procedure for constructing more complex prior classes, based on an adaptive choice of complexity level.

**Adaptive-combination of prior classes.** To address the challenge of setting the complexity level of the prior class, we introduce an adaptive (learnable) construction scheme for a prior class. Let $w_j(\boldsymbol{x}, t) : \mathbb{R}^d \times \mathbb{R}_+ \to \mathbb{R}$, $1 \le j \le k$, be learnable functions, which are part of the reconstruction model, i.e., $w_j(\cdot, t) \in \Psi_t$ and $w_j$ are normalized, i.e., $\sum_{j=1}^{k} w_j(\boldsymbol{x}, t) = 1$. The details of $w_j$ architecture are left to Section 5. We can construct a complex prior class by assigning simpler prior classes to different parts of the space, according to the weights $w_j$. For example, let us consider a *piece-wise* rigid deformation prior class defined as:

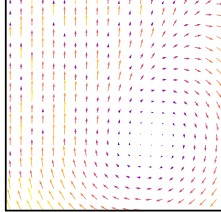

Figure 1: A vector field in $\mathcal{P}_V$.

$$\mathcal{P}_{IV} = \left\{ u_t \mid u(\boldsymbol{x}, t) = \sum_{j=1}^{k} w_j(\boldsymbol{x}, t) u_j(\boldsymbol{x}, t), u_j \in \mathcal{P}_{II} \text{ for } 1 \le j \le k \right\}. \tag{23}$$

In a similar manner, we can also combine $\mathcal{P}_I$ with rigid deformations and derive a prior class defined as:

$$\mathcal{P}_V = \left\{ u_t | u(\boldsymbol{x}, t) = \sum_{j=1}^{k} w_j(\boldsymbol{x}, t) u_j(\boldsymbol{x}, t), u_1 \in \mathcal{P}_I, u_j \in \mathcal{P}_{II} \text{ for } 2 \leq j \leq k \right\}. \tag{24}$$

Figure 1 illustrates an element in $\mathcal{P}_V$, with weights $w_j$ dividing the plane to a restricted up direction deformation above the diagonal, and a rigid deformation below the diagonal. Note that directly substituting an adaptive-combination prior class in 5 would no longer yield a linear problem. Therefore, we propose to use a linear problem that upper bounds the matching optimization problem of 5. For example, in the case of equation 9 with $\mathcal{P}_{IV}$, we can solve:

$$\min_{\{(\boldsymbol{A}_{jt}, \boldsymbol{b}_{jt})\}} \sum_{i=1}^{n} \sum_{j=1}^{k} w_j(\gamma_t^i, t) \left\| \boldsymbol{A}_{jt} \gamma_t^i + \boldsymbol{b}_{jt} - \frac{d}{dt} \gamma_t^i \right\|^2 \text{ s.t. } \boldsymbol{A}_{jt} = -\boldsymbol{A}_{jt}^T. \tag{25}$$

Using Jensen's inequality, it can be seen that 25 upper bounds the matching optimization from 5. Now, problem 25 can be solved efficiently, as it corresponds to a weighted least squares problem that is solvable in parallel for each $j \in [k]$, similarly to problem 17.

Lastly, incorporating $\mathcal{P}_{II}$ into equation 5 results in a non-standard least-squares problem with a constraint. The following lemma, with its proof provided in the Appendix, formulates the solutions for using $\mathcal{P}_{IV}$ in equation 5, covering problems 17 and 18 as a special case.

**Lemma 1.** *For the prior class $\mathcal{P}_{IV}$, the solutions $(\boldsymbol{A}_{jt}, \boldsymbol{b}_{jt})$ to the minimization problem 25 are given by,*

$$\begin{bmatrix} \mathrm{vech}(\boldsymbol{A}_{jt}) \\ \boldsymbol{b}_{jt} \end{bmatrix} = \boldsymbol{P}_{jt}^{-1} \begin{bmatrix} \mathrm{vec}(\dot{\boldsymbol{\Gamma}}_t^T \boldsymbol{W}_{jt} \boldsymbol{\Gamma}_{jt} - \boldsymbol{\Gamma}_{jt}^T \boldsymbol{W}_{jt} \dot{\boldsymbol{\Gamma}}_t) \\ \frac{1}{\mathbf{1}^T \boldsymbol{W}_{jt} \mathbf{1}} \mathbf{1}^T \boldsymbol{W}_{jt} \dot{\boldsymbol{\Gamma}}_t \end{bmatrix}$$

*where $\boldsymbol{\Gamma}_{jt} = \left[ \gamma_t^1, \cdots, \gamma_t^n \right]^T \in \mathbb{R}^{n \times d}$, $\dot{\boldsymbol{\Gamma}}_t = \left[ \frac{d}{dt} \gamma_t^1, \cdots, \frac{d}{dt} \gamma_t^n \right]^T \in \mathbb{R}^{n \times d}$, $\boldsymbol{W}_{jt} = \sum_{i=1}^{n} w_j(\gamma_t^i, t) \boldsymbol{e}_i \boldsymbol{e}_i^T$ with $\{\boldsymbol{e}_i\}$ as the standard basis in $\mathbb{R}^n$, $\mathrm{vech}(\boldsymbol{A}_{jt}) \in \mathbb{R}^{\frac{d(d-1)}{2}}$ denotes the half-vectorization of the anti-symmetric matrix $\boldsymbol{A}_{jt}$, and the matrix $\boldsymbol{P}_{jt}^{-1}$ depends solely on $\boldsymbol{\Gamma}_{jt}$, $\dot{\boldsymbol{\Gamma}}_t$, and $\boldsymbol{W}_{jt}$.*

*The solutions $(\boldsymbol{A}_{jt}, \boldsymbol{b}_{jt})$ to the minimization problem 5 with 10 are given by,*

$$\begin{bmatrix} \mathrm{vech}(\boldsymbol{A}_{jt}) \\ \boldsymbol{b}_{jt} \end{bmatrix} = \boldsymbol{P}_{jt}^{-1} \begin{bmatrix} \sum_{i=1}^{n} w_j(\boldsymbol{x}_i, t) \mathrm{vec}(s_i(\boldsymbol{x}_i [\boldsymbol{g}_t^i]^T - \boldsymbol{g}_t^i \boldsymbol{x}_i^T)) \\ \boldsymbol{G}_t^T \boldsymbol{W}_{jt} \boldsymbol{s} \end{bmatrix}$$

*where $\boldsymbol{g}_t^i = \left[ \nabla \psi_t(\boldsymbol{x}_i) \right]^T$, $\boldsymbol{G}_t = \left[ \boldsymbol{g}_t^1, \cdots, \boldsymbol{g}_t^n \right]^T \in \mathbb{R}^{n \times d}$, $s_i = \frac{\partial}{\partial t} \psi_t(\boldsymbol{x}_i)$, $\boldsymbol{s} = \left[ s_t^1, \cdots, s_t^n \right]^T \in \mathbb{R}^n$.*

## 5 IMPLEMENTATION DETAILS

In this section, we provide additional details about the dynamic image model $\Psi_t$ employed in this work, based on Gaussian Splatting (Kerbl et al., 2023). We provide an overview of this image model, followed by details about the dynamic model used in the experiments.

**Gaussian Splatting image model.** The Gaussian Splatting image model is parameterized by a collection of $n$ 3D Gaussians augmented with color and opacity parameters. That is, $\boldsymbol{\theta} = \left\{ \boldsymbol{\mu}^i, \boldsymbol{\Sigma}^i, \boldsymbol{c}^i, \alpha^i \right\}_{i=1}^{n}$ with $\boldsymbol{\mu}^i \in \mathbb{R}^3$ denoting the $i^{\text{th}}$ Gaussian mean, $\boldsymbol{\Sigma}^i \in \mathbb{R}^{3 \times 3}$ its covariance matrix, $\boldsymbol{c}^i \in \mathbb{R}^3$ its color, and $\alpha^i \in \mathbb{R}$ its opacity. To render an image, the 3D Gaussians are projected to the image plane to form a collection of 2D Gaussians parameterized by $\left\{ \boldsymbol{\mu}_{\text{2D}}^i, \boldsymbol{\Sigma}_{\text{2D}}^i \right\}$. Given $K, E$ denoting the intrinsic and extrinsic camera transformations, the image plane Gaussians parameters are calculated using the point rendering formula:

$$\boldsymbol{\mu}_{\text{2D}}^i = K \frac{E \boldsymbol{\mu}^i}{(E \boldsymbol{\mu}^i)_{\boldsymbol{z}}}, \boldsymbol{\Sigma}_{\text{2D}}^i = J E \boldsymbol{\Sigma}^i E^T J^T \tag{26}$$

where $J$ denotes the Jacobian of the affine transformation of 26. Lastly, an image pixel $I(\boldsymbol{p})$ is obtained by alpha-blending the ordered by depth visible Gaussians:

$$I(\boldsymbol{p}) = \sum_{i=1}^{n} \boldsymbol{c}^i \alpha^i \sigma^i(\boldsymbol{p}) \prod_{j=1}^{i-1} \left( 1 - \alpha^j \sigma^j(\boldsymbol{p}) \right), \tag{27}$$

where $\sigma^i(\boldsymbol{p}) = \exp \left( -\frac{1}{2} \left( \boldsymbol{p} - \boldsymbol{\mu}_{\text{2D}}^i \right)^T \left( \boldsymbol{\Sigma}_{\text{2D}}^i \right)^{-1} \left( \boldsymbol{p} - \boldsymbol{\mu}_{\text{2D}}^i \right) \right)$.

**Dynamic image model.** We utilize the Gaussian Splatting image model to construct our dynamic model as:

$$\Psi_t = \left\{ \boldsymbol{\mu}^i + \boldsymbol{\mu}^i(t), \boldsymbol{\Sigma} + \boldsymbol{\Sigma}^i(t), \boldsymbol{c}^i, \alpha^i, w_{ij}(t) \right\}_{i=1}^n, \tag{28}$$

where $\boldsymbol{\mu}^i(t) = f_{\boldsymbol{\mu}}(\boldsymbol{\mu}^i, t)$, $\boldsymbol{\Sigma}^i(t) = f_{\boldsymbol{\Sigma}}(\boldsymbol{\mu}^i, t)$, $w_{ij}(t) = e_j^T \mathrm{softmax}(f_w(\boldsymbol{\mu}^i + \boldsymbol{\mu}^i(t), \boldsymbol{\mu}^i, t))$. We follow Yang et al. (2023) and each of the functions: $f_{\boldsymbol{\mu}} : \mathbb{R}^3 \times \mathbb{R} \to \mathbb{R}^3$, $f_{\boldsymbol{\Sigma}} : \mathbb{R}^3 \times \mathbb{R} \to \mathbb{R}^6$, $f_w : \mathbb{R}^3 \times \mathbb{R}^3 \times \mathbb{R} \to \mathbb{R}^k$ is a Multilayer perceptron (MLP). For more details regarding the MLP architectures, we refer the reader to the Appendix. Note that the model element $w_{ij}(t)$ is only relevant to instances where the adaptive-combination prior class is assumed. Lastly, in our experiments we apply the ReMatching loss for $\boldsymbol{\mu}^i + \boldsymbol{\mu}^i(t)$, and for time-dependent rendered images $I_t$.

**Training details.** We follow the training protocol of (Yang et al., 2023). We initialize the model using $n = 100K$ 3D Gaussians. Training is done for 40K iterations, where for the first 3K iterations, only $\left\{ \boldsymbol{\mu}^i, \boldsymbol{\Sigma}^i, \boldsymbol{c}^i, \alpha^i \right\}_{i=1}^n$ are optimized. In instances where the adaptive-combination prior class is applied, we supplement the ReMatching optimization objective with an entropy loss on the weights $w_{ij}$ as follows:

$$L_{\mathrm{entropy}} = \frac{1}{k} \sum_{j=1}^k \frac{1}{n} \sum_{i=1}^n w_{ij} \log \left( \frac{1}{n} \sum_{i=1}^n w_{ij} \right). \tag{29}$$

Lastly, for all the experiments considered in this work, we set the ReMatching loss weight $\lambda = 0.001$.

## 6 EXPERIMENTS

We evaluate the ReMatching framework on benchmarks involving synthetic and real-world video captures of deforming scenes. For quantitative analysis in both cases, we report the PSNR, SSIM (Wang et al., 2004)and LPIPS (Zhang et al., 2018) metrics. Additional evaluations, including experiments on more synthetic and real-world datasets, hyperparameter ablation, and the framework's applicability to an alternative dynamic image model, are provided in the Appendix.

**D-NeRF synthetic.** D-NeRF dataset (Pumarola et al., 2021) comprises of 8 scenes, each consisting from 100 to 200 frames, hence providing a dense multi-view coverage of the scene. We follow D-NeRF's evaluation protocol and use the same train/validation/test split at $800 \times 800$ image resolution with a black background. In terms of baseline methods, we consider recent state-of-the-art dynamic models, including Deformable 3D Gaussians (D3G) (Yang et al., 2023), 3D Geometry-aware Deformable Gaussians (GA3D) (Lu et al., 2024), Neural Parametric Gaussians (NPG) (Das et al., 2024), and K-Planes (Fridovich-Keil et al., 2023). Note that some of these baselines incorporate prior regularization losses such as local rigidity and smoothness to their optimization procedure. Table 1 summarizes the average image quality results for unseen frames in each scene. We include two variants of our framework: i) Using the divergence-free prior $\mathcal{P}_{III}$; and ii) Using the adaptive-combination prior class $\mathcal{P}_{IV}$ or the class $\mathcal{P}_V$ specifically for scenes that include a floor component. Figure 2 provides a qualitative comparison of rendered test frames, highlighting the improvements of our approach, which: i) produces plausible reconstructions that avoid unrealistic distortions, e.g., the human fingers in the jumping jacks scene; ii) reduces rendering artifacts of extraneous parts, especially in moving parts such as the leg in the T-Rex scene.

| Method | Bouncing Balls | | | Hell Warrior | | | Hook | | | JumpingJacks | | |
|---|---|---|---|---|---|---|---|---|---|---|---|---|
| | LPIPS ↓ | PSNR ↑ | SSIM ↑ | LPIPS ↓ | PSNR ↑ | SSIM ↑ | LPIPS ↓ | PSNR ↑ | SSIM ↑ | LPIPS ↓ | PSNR ↑ | SSIM ↑ |
| K-Planes (Fridovich-Keil et al., 2023) | 0.0242 | 37.78 | 0.9929 | 0.1074 | 32.57 | 0.9316 | 0.0655 | 29.46 | 0.9481 | 0.0417 | 31.73 | 0.9715 |
| NPG (Das et al., 2024) | | | | 0.0537 | 38.68 | 0.9780 | 0.0460 | 33.39 | 0.9735 | 0.0345 | 33.97 | 0.9828 |
| GA3D (Lu et al., 2024) | 0.0093 | 40.76 | 0.9950 | 0.0210 | 41.30 | 0.9871 | 0.0124 | 37.78 | 0.9887 | 0.0121 | 37.00 | 0.9887 |
| D3G (Yang et al., 2023) | 0.0089 | 41.52 | 0.9978 | 0.0261 | 41.28 | 0.9928 | 0.0165 | 37.03 | 0.9906 | 0.0137 | 37.59 | 0.9930 |
| Ours ($\mathcal{P}_{III}$) | 0.0087 | 41.84 | 0.9979 | 0.0244 | 41.59 | 0.9932 | 0.0161 | 37.19 | 0.9909 | 0.0134 | 37.72 | 0.9931 |
| Ours ($\mathcal{P}_{IV}$ or $\mathcal{P}_V$) | 0.0089 | 41.61 | 0.9978 | 0.0245 | 41.69 | 0.9977 | 0.0158 | 37.39 | 0.9911 | 0.0131 | 38.01 | 0.9934 |

| Method | Lego | | | Mutant | | | Stand Up | | | T-Rex | | |
|---|---|---|---|---|---|---|---|---|---|---|---|---|
| | LPIPS ↓ | PSNR ↑ | SSIM ↑ | LPIPS ↓ | PSNR ↑ | SSIM ↑ | LPIPS ↓ | PSNR ↑ | SSIM ↑ | LPIPS ↓ | PSNR ↑ | SSIM ↑ |
| K-Planes (Fridovich-Keil et al., 2023) | 0.0472 | 25.15 | 0.9431 | 0.0215 | 35.30 | 0.9825 | 0.0211 | 36.55 | 0.9831 | 0.0284 | 30.41 | 0.9778 |
| NPG (Das et al., 2024) | 0.0716 | 24.63 | 0.9312 | 0.0311 | 36.02 | 0.9840 | 0.0257 | 38.20 | 0.9889 | 0.0310 | 32.10 | 0.9959 |
| GA3D (Lu et al., 2024) | 0.0446 | 24.87 | 0.9420 | 0.0050 | 42.39 | 0.9951 | 0.0062 | 43.96 | 0.9948 | 0.0100 | 37.70 | 0.9929 |
| D3G (Yang et al., 2023) | 0.0453 | 24.93 | 0.9537 | 0.0066 | 42.09 | 0.9966 | 0.0083 | 43.85 | 0.9970 | 0.0105 | 37.89 | 0.9956 |
| Ours ($\mathcal{P}_{III}$) | 0.0503 | 24.89 | 0.9522 | 0.0067 | 42.13 | 0.9966 | 0.0085 | 43.99 | 0.9969 | 0.0105 | 38.07 | 0.9958 |
| Ours ($\mathcal{P}_{IV}$ or $\mathcal{P}_V$) | 0.0456 | 24.95 | 0.9537 | 0.0065 | 42.40 | 0.9968 | 0.0081 | 44.31 | 0.9971 | 0.0103 | 38.38 | 0.9961 |

Table 1: Image quality evaluation on unseen frames for the D-NeRF dataset (Pumarola et al., 2021).

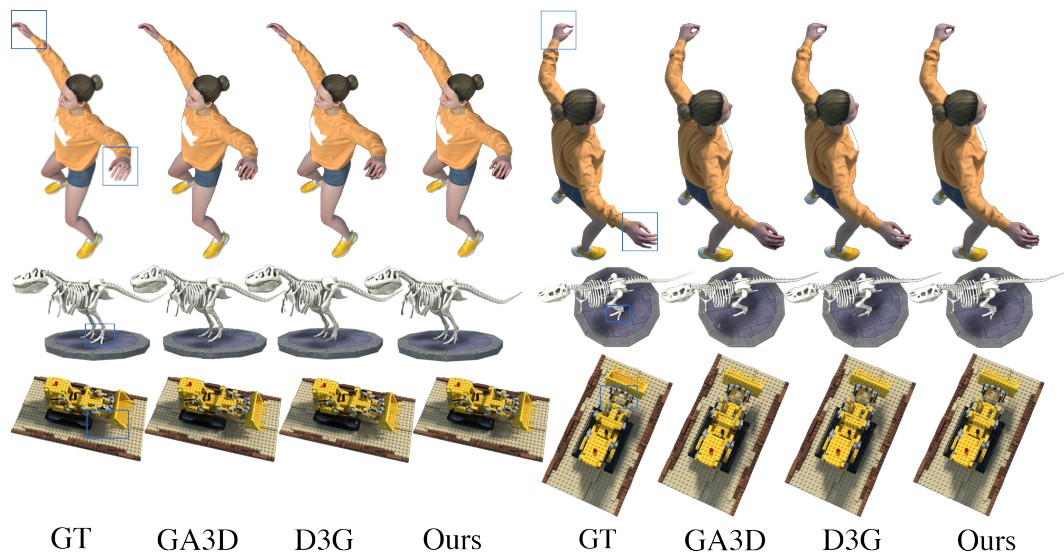

| GT | GA3D | D3G | Ours | GT | GA3D | D3G | Ours |

Figure 2: Qualitative comparison of baselines and our model on the D-NeRF dataset (Pumarola et al., 2021). We note that our framework consistently produces high fidelity reconstructions, accurately capturing fine-grained details, as highlighted in the blue boxes.

**HyperNeRF real-world.** The HyperNeRF dataset (Park et al., 2021b) consists of real-world videos capturing a diverse set of human activities involving interactions with common objects. We follow the evaluation protocol provided with the dataset, and use the same train/test split. In table 2 we report image quality results for unseen frames on 5 scenes from the dataset: Slice Banana, Chicken, Lemon, Torch, and Split Cookie. Figure 3 shows qualitative comparison to the baseline D3G (Yang et al., 2023). Our approach demonstrates similar types of improvements as noticed in the synthetic case providing more realistic reconstructions, especially in areas involving deforming parts.

| Scene | | LPIPS ↓ | PSNR ↑ | SSIM ↑ |
|---|---|---|---|---|
| | GA3D | 0.4160 | 25.34 | 0.6722 |
| | D3G | 0.3692 | 24.87 | 0.7935 |
| Slice Banana | Ours ($\mathcal{P}_{III}$) | 0.3829 | 25.08 | 0.7992 |
| | Ours ($\mathcal{P}_{IV}$) | 0.3673 | 25.28 | 0.8025 |
| | GA3D | 0.4721 | 25.13 | 0.7555 |
| | D3G | 0.3030 | 26.66 | 0.8813 |
| Chicken | Ours ($\mathcal{P}_{III}$) | 0.2987 | 26.74 | 0.8836 |
| | Ours ($\mathcal{P}_{IV}$) | 0.3044 | 26.80 | 0.8835 |
| | GA3D | 0.3252 | 28.37 | 0.7596 |
| | D3G | 0.2858 | 28.65 | 0.8873 |
| Lemon | Ours ($\mathcal{P}_{III}$) | 0.2760 | 27.91 | 0.8842 |
| | Ours ($\mathcal{P}_{IV}$) | 0.2675 | 28.30 | 0.8883 |
| | GA3D | 0.3278 | 23.79 | 0.8174 |
| | D3G | 0.2340 | 25.41 | 0.9207 |
| Torch | Ours ($\mathcal{P}_{III}$) | 0.2221 | 26.00 | 0.9251 |
| | Ours ($\mathcal{P}_{IV}$) | 0.2260 | 25.62 | 0.9229 |
| | GA3D | 0.1144 | 32.28 | 0.9290 |
| | D3G | 0.0971 | 32.61 | 0.9657 |
| Split Cookie | Ours ($\mathcal{P}_{III}$) | 0.1097 | 31.31 | 0.9600 |
| | Ours ($\mathcal{P}_{IV}$) | 0.0937 | 32.67 | 0.9667 |

Table 2: Unseen frames evaluation for the Hyper-NeRF dataset (Park et al., 2021b).

**ReMatching time-dependent image.** In this experiment we validate the applicability of the ReMatching loss for controlling model solutions via rendered images. To that end, we apply our framework with the $\mathcal{P}_{III}$ prior class to the Jumping Jacks scene from D-NeRF on a single specific front view through time. The qualitative comparison to D3G (Yang et al., 2023), as shown in the Appendix, supports the benefits of prior integration in this case as well, demonstrating more plausible reconstructions in areas involving moving parts.

**Adaptive-combination prior class.** Employing the adaptive-combination prior classes $\mathcal{P}_{IV}$ and $\mathcal{P}_V$ with learnable parts assignments $\{w_{ij}\}$ raises the question of whether the learning process successfully produced assignments $\{w_{ij}\}$ that align with the scene segmentation based on its deforming parts. Figure 4 shows our results for test frames from the Bouncing-Balls and Lego synthetic scenes (left), and the Chicken real-world scene (right). For comparison, we include the results of the Segment Anything Model (SAM) (Kirillov et al., 2023), which are mostly influenced by color variations. Consequently, SAM often over-segments the scene or incorrectly merges independently moving parts. See the supplementary material for additional segmentation results.

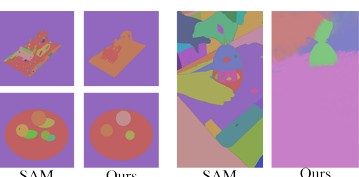

Figure 4: Part assignments for the adaptive-combination prior class.

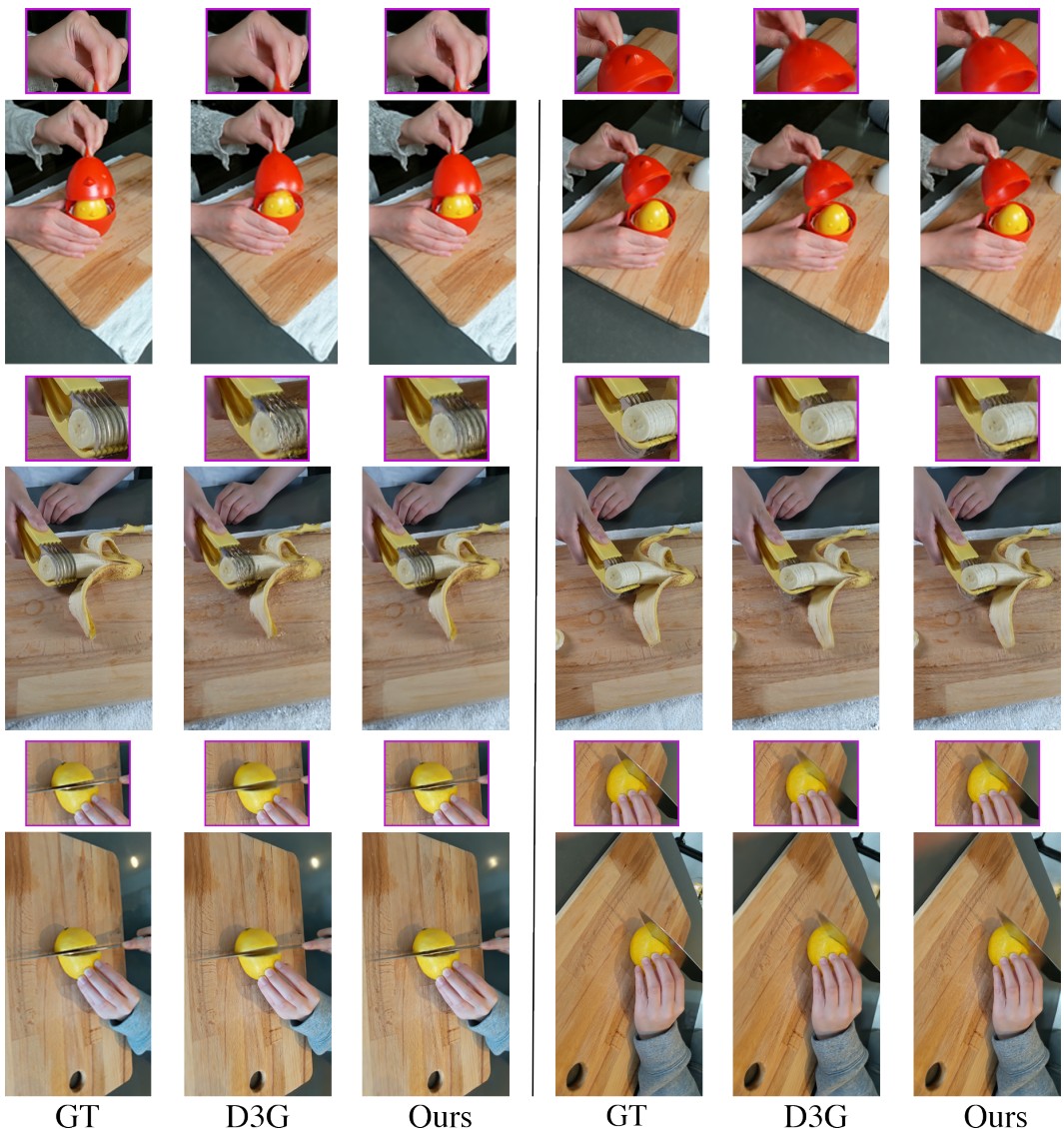

GT          D3G          Ours          GT          D3G          Ours

Figure 3: Qualitative comparison of our method to D3G (Yang et al., 2023) on the HyperNeRF dataset (Park et al., 2021b). Our framework yields more accurate reconstructions, in particular around moving parts.

## 7   CONCLUSIONS

We presented the ReMatching framework for integrating prior deformation classes into dynamic reconstruction models. In addition to offering useful constructions for velocity-field-based prior classes, a key focus of this work was the development of the ReMatching loss. This loss function optimizes for solutions that remain as close as possible to the desired prior class, rather than strictly enforcing membership, thereby minimizing the tradeoff between fidelity and the advantages of using priors. Our experimental results validate this approach, showing that ReMatching solutions successfully adhere to the desired prior while achieving high-fidelity reconstruction. We believe that the generality with which the framework was formulated will enable broader applicability and that future advancements in this framework could be relevant to a wide range of dynamic reconstruction models. An interesting avenue for future research is the development of velocity-field-based prior classes emerging from video generative models, potentially using our ReMatching formulation for time-dependent image pixels color. Another promising direction is the design of richer prior classes to handle more complex physical phenomena, such as ones including liquids and gases.

ACKNOWLEDGMENTS

The authors would like to thank Jonathan Lorraine and Heli Ben-Hamu for their insightful discussions and valuable comments.

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

# 8 APPENDIX

## 8.1 PROOFS

### 8.1.1 PROOF OF LEMMA 1

*Proof.* (Lemma 1)

Let $\mathbf{\Gamma}_t, \dot{\mathbf{\Gamma}}_t, \mathbf{W}_{jt}$ be given. Without loss of generality, we show the proof only for individuals $j \in [k]$ and $t$. So in order to ease the notation, in what follows we omit the subscripts $t$ and $j$. Let $\mathbf{A} \in \mathbb{R}^{d \times d}$ and $\mathbf{b} \in \mathbb{R}^d$, and define $[w_1, \cdots, w_n]^T = \mathbf{W}\mathbf{1}$. First, we show that $\sum_{i=1}^n w_i \left\| \mathbf{A}\gamma^i + \mathbf{b} - \dot{\gamma}^i \right\|^2$ can be reformulated as a weighted norm-squared minimization problem in $\mathbf{A}$ and $\mathbf{b}$. That is,

$$\sum_{i=1}^n w_i \left\| \mathbf{A}\gamma^i + \mathbf{b} - \dot{\gamma}^i \right\|^2 = \sum_{i=1}^n \mathrm{tr}\mathbf{A}\sqrt{w_i}\gamma^i\sqrt{w_i}\gamma^{i^T}\mathbf{A}^T - \tag{30}$$

$$2\mathrm{tr}\sqrt{w_i}(\dot{\gamma}^i - \mathbf{b})\sqrt{w_i}\gamma^i\mathbf{A}^T + \mathrm{tr}\sqrt{w_i}(\dot{\gamma}^i - \mathbf{b})\sqrt{w_i}(\dot{\gamma}^i - \mathbf{b})^T \tag{31}$$

$$= \mathrm{tr}\mathbf{A}\mathbf{\Gamma}^T\mathbf{W}\mathbf{\Gamma}\mathbf{A}^T - 2\mathrm{tr}(\dot{\mathbf{\Gamma}} - \mathbf{1}\mathbf{b}^T)^T\mathbf{W}\mathbf{\Gamma}\mathbf{A}^T + \tag{32}$$

$$\mathrm{tr}(\dot{\mathbf{\Gamma}} - \mathbf{1}\mathbf{b}^T)^T\mathbf{W}(\dot{\mathbf{\Gamma}} - \mathbf{1}\mathbf{b}^T) \tag{33}$$

$$= \left\| \sqrt{\mathbf{W}}\left(\mathbf{\Gamma}\mathbf{A}^T - (\dot{\mathbf{\Gamma}} - \mathbf{1}\mathbf{b}^T)\right) \right\|^2. \tag{34}$$

Next, we consider the following optimization problem:

$$\min_{\mathbf{A}, \mathbf{b}} \left\| \sqrt{\mathbf{W}}\left(\mathbf{\Gamma}\mathbf{A}^T - (\dot{\mathbf{\Gamma}} - \mathbf{1}\mathbf{b}^T)\right) \right\|^2 \text{ s.t. } \mathbf{A} = -\mathbf{A}^T. \tag{35}$$

Use the fact that $\boldsymbol{A} = -\boldsymbol{A}^T$ to define the following Lagrangian,

$$\mathcal{L}(\boldsymbol{A}, \boldsymbol{b}, \Lambda) = \left\| \sqrt{\boldsymbol{W}} \left( \boldsymbol{\Gamma}\boldsymbol{A} - \mathbf{1}\boldsymbol{b}^T + \dot{\boldsymbol{\Gamma}} \right) \right\|^2 + \mathrm{tr}\Lambda^T \left( \boldsymbol{A} + \boldsymbol{A}^T \right). \tag{36}$$

Then,

$$\frac{\partial \mathcal{L}}{\partial \boldsymbol{A}} = 2\boldsymbol{\Gamma}^T \boldsymbol{W} \left( \boldsymbol{\Gamma}\boldsymbol{A} - \mathbf{1}\boldsymbol{b}^T + \dot{\boldsymbol{\Gamma}} \right) + \Lambda + \Lambda^T.$$

Thus, $\frac{\partial \mathcal{L}}{\partial \boldsymbol{A}} = 0$ yields that $\boldsymbol{\Gamma}^T \boldsymbol{W} (\boldsymbol{\Gamma}\boldsymbol{A} - \mathbf{1}\boldsymbol{b}^T + \dot{\boldsymbol{\Gamma}})$ is symmetric. Then, using again the fact that $\boldsymbol{A} = -\boldsymbol{A}^T$, we get that,

$$\boldsymbol{\Gamma}^T \boldsymbol{W}\boldsymbol{\Gamma}\boldsymbol{A} + \boldsymbol{A}\boldsymbol{\Gamma}^T \boldsymbol{W}\boldsymbol{\Gamma} + \boldsymbol{b}\mathbf{1}^T \boldsymbol{W}\boldsymbol{\Gamma} - \boldsymbol{\Gamma}^T \boldsymbol{W}\mathbf{1}\boldsymbol{b}^T = \dot{\boldsymbol{\Gamma}}^T \boldsymbol{W}\boldsymbol{\Gamma} - \boldsymbol{\Gamma}^T \boldsymbol{W}\dot{\boldsymbol{\Gamma}}. \tag{37}$$

Now, taking the derivative w.r.t. to $\boldsymbol{b}$ gives,

$$\frac{\partial \mathcal{L}}{\partial \boldsymbol{b}} = -2\mathbf{1}^T \boldsymbol{W} \left( \boldsymbol{\Gamma}\boldsymbol{A} - \mathbf{1}\boldsymbol{b}^T + \dot{\boldsymbol{\Gamma}} \right)$$

and, $\frac{\partial \mathcal{L}}{\partial \boldsymbol{b}} = 0$, yields that,

$$-\widehat{\boldsymbol{w}}^T \boldsymbol{\Gamma}\boldsymbol{A} + \boldsymbol{b}^T = \widehat{\boldsymbol{w}}^T \dot{\boldsymbol{\Gamma}}, \tag{38}$$

where $\widehat{\boldsymbol{w}} = \frac{\boldsymbol{W}\mathbf{1}}{\mathbf{1}^T \boldsymbol{W}\mathbf{1}}$. Vectorizing the LHS of 37 gives,

$$\left( I_d \otimes \boldsymbol{\Gamma}^T \boldsymbol{W}\boldsymbol{\Gamma} + \boldsymbol{\Gamma}^T \boldsymbol{W}\boldsymbol{\Gamma} \otimes I_d \right) D_d \mathrm{vech}(\boldsymbol{A}) + \left( \boldsymbol{\Gamma}^T \boldsymbol{W}\mathbf{1} \otimes I_d - I_d \otimes \boldsymbol{\Gamma}^T \boldsymbol{W}\mathbf{1} \right) \boldsymbol{b} \tag{39}$$

where $D_d$ is the duplication matrix transforming $\mathrm{vech}(\boldsymbol{A})$ to $\mathrm{vec}(\boldsymbol{A})$, with $\mathrm{vech}(\boldsymbol{A})$ denoting the half-vectorization of the anti-symmetric matrix $\boldsymbol{A}$. Similarly, vectorizing the LHS of 38 yields,

$$-\frac{1}{\mathbf{1}^T \boldsymbol{W}\mathbf{1}} \left( I_d \otimes \mathbf{1}^T \boldsymbol{W}\boldsymbol{\Gamma} \right) D_d \mathrm{vech}(\boldsymbol{A}) + \boldsymbol{b}. \tag{40}$$

Based on 39 and 40, we can define the following block matrix:

$$\boldsymbol{P} = \left[ \begin{array}{c|c} \boldsymbol{Q} = \boldsymbol{Q}'D_d & \boldsymbol{R} = \boldsymbol{\Gamma}^T \boldsymbol{W}\mathbf{1} \otimes I_d - I_d \otimes \boldsymbol{\Gamma}^T \boldsymbol{W}\mathbf{1} \\ \hline \boldsymbol{S} = \boldsymbol{S}'D_d & \boldsymbol{T} = I_d \end{array} \right] \tag{41}$$

where $\boldsymbol{Q}' = I_d \otimes \boldsymbol{\Gamma}^T \boldsymbol{W}\boldsymbol{\Gamma} + \boldsymbol{\Gamma}^T \boldsymbol{W}\boldsymbol{\Gamma} \otimes I_d$, and, $\boldsymbol{S}' = -\frac{1}{\mathbf{1}^T \boldsymbol{W}\mathbf{1}} \left( I_d \otimes \mathbf{1}^T \boldsymbol{W}\boldsymbol{\Gamma} \right)$. Then, let,

$$\boldsymbol{U} = \left( \boldsymbol{Q} - \boldsymbol{R}\boldsymbol{T}^{-1}\boldsymbol{S} \right)^{-1} = L_d \left( \boldsymbol{Q}' - \boldsymbol{R}\boldsymbol{S}' \right)^{-1} \tag{42}$$

where $L_d$ is the matrix satisfying $D_d L_d = I_{d^2}$. Consequently,

$$\boldsymbol{P}^{-1} = \left[ \begin{array}{c|c} \boldsymbol{U} & -\boldsymbol{U}\boldsymbol{R} \\ \hline -\boldsymbol{S}' \left( \boldsymbol{Q}' - \boldsymbol{R}\boldsymbol{S}' \right)^{-1} & I_d + \boldsymbol{S}' \left( \boldsymbol{Q}' - \boldsymbol{R}\boldsymbol{S}' \right)^{-1} \boldsymbol{R} \end{array} \right] \tag{43}$$

and,

$$\left[ \begin{array}{c} \mathrm{vech}(\boldsymbol{A}) \\ \boldsymbol{b} \end{array} \right] = \boldsymbol{P}^{-1} \left[ \begin{array}{c} \mathrm{vec}(\dot{\boldsymbol{\Gamma}}^T \boldsymbol{W}\boldsymbol{\Gamma} - \boldsymbol{\Gamma}^T \boldsymbol{W}\dot{\boldsymbol{\Gamma}}) \\ \widehat{\boldsymbol{w}}^T \dot{\boldsymbol{\Gamma}} \end{array} \right]. \tag{44}$$

$\square$

Now, for the second part of the lemma. Let $\boldsymbol{g}_i = \left[ \nabla \psi_t(\boldsymbol{x}_i) \right]^T$, $s_i = \frac{\partial}{\partial t} \psi_t(\boldsymbol{x}_i)$. Consider the following energy,

$$L = \sum_{i=1}^{n} w_i \left( \boldsymbol{g}_i^T \left( A\boldsymbol{x}_i + \boldsymbol{b} \right) + s_i \right)^2. \tag{45}$$

Note that,

$$\boldsymbol{g}_i^T A\boldsymbol{x}_i = \boldsymbol{y}_i^T \boldsymbol{a} \tag{46}$$

where $\boldsymbol{a} := \mathrm{vec}(A)$, and $\boldsymbol{y}_i := \boldsymbol{x}_i \otimes \boldsymbol{g}_i$. Then,

$$L = \sum_{i=1}^{n} w_i \left( \boldsymbol{a}^T \boldsymbol{y}_i \boldsymbol{y}_i^T \boldsymbol{a} + \boldsymbol{b}^T \boldsymbol{g}_i \boldsymbol{g}_i^T \boldsymbol{b} + 2\boldsymbol{a}^T \boldsymbol{y}_i \boldsymbol{g}_i^T \boldsymbol{b} + 2\boldsymbol{g}_i^T s_i \boldsymbol{b} + 2s_i \boldsymbol{a}^T \boldsymbol{y}_i + s_i^2 \right). \tag{47}$$

Define the Lagrangian,

$$\mathcal{L}(\boldsymbol{a}, \boldsymbol{b}, \lambda) = \boldsymbol{a}^T \sum_i \boldsymbol{y}_i w_i \boldsymbol{y}_i^T \boldsymbol{a} + \boldsymbol{b}^T \boldsymbol{G}^T \boldsymbol{W} \boldsymbol{G} \boldsymbol{b} + 2\boldsymbol{a}^T \sum_i \boldsymbol{y}_i w_i \boldsymbol{g}_i^T \boldsymbol{b} + \tag{48}$$

$$2\boldsymbol{s}^T \boldsymbol{W} \boldsymbol{G} \boldsymbol{b} + 2\boldsymbol{a}^T \sum_i w_i \boldsymbol{y}_i s_i + \boldsymbol{t}^T \boldsymbol{W} \boldsymbol{t} + \lambda^T (\boldsymbol{a} + P\boldsymbol{a}) \tag{49}$$

where $P$ is the permutation matrix s.t. $\text{vec}(A^T) = P\boldsymbol{a}$.

Then,

$$\frac{\partial \mathcal{L}}{\partial \boldsymbol{a}} = 2 \sum_i w_i \boldsymbol{y}_i \boldsymbol{y}_i^T \boldsymbol{a} + 2 \sum_i w_i \boldsymbol{y}_i \boldsymbol{g}_i^T \boldsymbol{b} + 2 \sum_i w_i \boldsymbol{y}_i s_i + \lambda + P\lambda \tag{50}$$

Equating the above to $0$ and unvectorizing it, yields the following matrix equation,

$$\sum_{i=1}^n w_i (\boldsymbol{g}_i \boldsymbol{g}_i^T A \boldsymbol{x}_i \boldsymbol{x}_i^T + \boldsymbol{g}_i \boldsymbol{g}_i^T \boldsymbol{b} \boldsymbol{x}_i^T + s_i \boldsymbol{g}_i \boldsymbol{x}_i^T) = \frac{1}{2}(\boldsymbol{\Lambda} + \boldsymbol{\Lambda}^T), \tag{51}$$

yielding that the LHS is a symmetric matrix. Therefore,

$$\sum_{i=1}^n w_i (\boldsymbol{g}_i \boldsymbol{g}_i^T A \boldsymbol{x}_i \boldsymbol{x}_i^T + \boldsymbol{x}_i \boldsymbol{g}_i^T \boldsymbol{b} \boldsymbol{x}_i^T + s_i \boldsymbol{g}_i \boldsymbol{x}_i^T) = \sum_{i=1}^n w_i (\boldsymbol{x}_i \boldsymbol{x}_i^T A^T \boldsymbol{g}_i \boldsymbol{g}_i^T + \boldsymbol{x}_i \boldsymbol{b}^T \boldsymbol{g}_i \boldsymbol{g}_i^T + s_i \boldsymbol{x}_i \boldsymbol{g}_i^T). \tag{52}$$

Rearranging the above and half-vectorizing both sides yields that,

$$\sum_{i=1}^n w_i (\boldsymbol{x}_i \boldsymbol{x}_i^T \otimes \boldsymbol{g}_i \boldsymbol{g}_i^T + \boldsymbol{g}_i \boldsymbol{g}_i^T \otimes \boldsymbol{x}_i \boldsymbol{x}_i^T) D_d \text{vech}(\boldsymbol{A}) + w_i (\boldsymbol{x}_i \otimes \boldsymbol{g}_i \boldsymbol{g}_i^T - \boldsymbol{g}_i \boldsymbol{g}_i^T \otimes \boldsymbol{x}_i) \boldsymbol{b} = \tag{53}$$

$$\sum_{i=1}^n w_i \text{vec}(s_i (\boldsymbol{x}_i \boldsymbol{g}_i^T - \boldsymbol{g}_i \boldsymbol{x}_i^T)). \tag{54}$$

Now,

$$\frac{\partial \mathcal{L}}{\partial \boldsymbol{b}} = 0 \tag{55}$$

yields that,

$$\boldsymbol{G}^T \boldsymbol{W} \boldsymbol{G} \boldsymbol{b} + \sum_i w_i \boldsymbol{g}_i \boldsymbol{y}_i^T D_d \text{vech}(\boldsymbol{A}) = -\boldsymbol{G}^T \boldsymbol{W} \boldsymbol{s}. \tag{56}$$

Therefore,

$$P = \left[ \begin{array}{c|c} \boldsymbol{Q} = \boldsymbol{Q}' D_d & \boldsymbol{R} = \sum_{i=1}^n w_i (\boldsymbol{x}_i \otimes \boldsymbol{g}_i \boldsymbol{g}_i^T - \boldsymbol{g}_i \boldsymbol{g}_i^T \otimes \boldsymbol{x}_i) \\ \hline \boldsymbol{S} = \boldsymbol{S}' D_d & \boldsymbol{T} = \boldsymbol{G}^T \boldsymbol{W} \boldsymbol{G} \end{array} \right], \tag{57}$$

where $\boldsymbol{S}' = \sum_{i=1}^n w_i \boldsymbol{g}_i \boldsymbol{y}_i^T$, and $\boldsymbol{Q}' = \sum_{i=1}^n w_i (\boldsymbol{x}_i \boldsymbol{x}_i^T \otimes \boldsymbol{g}_i \boldsymbol{g}_i^T + \boldsymbol{g}_i \boldsymbol{g}_i^T \otimes \boldsymbol{x}_i \boldsymbol{x}_i^T)$. Then, let,

$$\boldsymbol{U} = (\boldsymbol{Q} - \boldsymbol{R} \boldsymbol{T}^{-1} \boldsymbol{S})^{-1} = L_d (\boldsymbol{Q}' - \boldsymbol{R} \boldsymbol{T}^{-1} \boldsymbol{S}')^{-1} \tag{58}$$

where $L_d$ is the matrix that satisfies $D_d L_d = I_{d^2}$. Consequently,

$$P^{-1} = \left[ \begin{array}{c|c} \boldsymbol{U} & -\boldsymbol{U} \boldsymbol{R} \boldsymbol{T}^{-1} \\ \hline -\boldsymbol{T}^{-1} \boldsymbol{S}' (\boldsymbol{Q}' - \boldsymbol{R} \boldsymbol{T}^{-1} \boldsymbol{S}')^{-1} & \boldsymbol{T}^{-1} + \boldsymbol{T}^{-1} \boldsymbol{S}' (\boldsymbol{Q}' - \boldsymbol{R} \boldsymbol{T}^{-1} \boldsymbol{S}')^{-1} \boldsymbol{R} \boldsymbol{T}^{-1} \end{array} \right]. \tag{59}$$

### 8.1.2 CONTINUITY EQUATION CONSTRAINT DERIVATION FOR $V = \mathbb{R}^{n \times d}$

In the main text, we stated that in the case when $V = \mathbb{R}^{n \times d}$, i.e., $\psi_t = (\gamma_t^1, \cdots, \gamma_t^n)^T$, equation 6 becomes:

$$\rho(u_t, \psi_t) = \sum_{i=1}^n \left\| u_t(\gamma_t^i) - \frac{d}{dt} \gamma_t^i \right\|^2. \tag{60}$$

To see this formally, let $\delta(\boldsymbol{x} - \boldsymbol{a})$ denote the Dirac delta generalized function concentrated around $\boldsymbol{a}$, satisfying

$$\delta(\boldsymbol{x} - \boldsymbol{a}) = 0, \forall \boldsymbol{x} \neq \boldsymbol{a}, \tag{61}$$

and,

$$\int \phi(\boldsymbol{x})\delta(\boldsymbol{x} - \boldsymbol{a})d\boldsymbol{x} = \phi(\boldsymbol{a}), \tag{62}$$

for any test function $\phi$. Consider $\psi_t(\boldsymbol{x}) = \sum_{i=1}^{n} \psi_t^i(\boldsymbol{x})$, where $\psi_t^i(\boldsymbol{x}) = \delta(\boldsymbol{x} - \gamma_t^i)$. Note that under this definition of $\psi_t$, $V$ is in fact the space of generalized functions. Then,

$$\frac{\partial}{\partial t}\psi_t^i = \left\langle \nabla\delta(\boldsymbol{x} - \gamma_t^i), -\frac{d}{dt}\gamma_t^i \right\rangle, \tag{63}$$

and, using the chain rule as applied in the simplification of equation 10, we have that,

$$\operatorname{div}\psi_t^i u(\boldsymbol{x}) = \left\langle \nabla\delta(\boldsymbol{x} - \gamma_t^i), u(\boldsymbol{x}) \right\rangle + \delta(\boldsymbol{x} - \gamma_t^i)\operatorname{div}(u(\boldsymbol{x})). \tag{64}$$

Substituting these computations in the continuity equation 4, yields that,

$$0 = \int \left| \frac{\partial}{\partial t}\psi_t(\boldsymbol{x}) + \operatorname{div}\left(\psi_t(\boldsymbol{x})v_t\left(\boldsymbol{x}\right)\right) \right| d\boldsymbol{x} \geq \tag{65}$$

$$\left| \int \frac{\partial}{\partial t}\psi_t(\boldsymbol{x}) + \operatorname{div}\left(\psi_t(\boldsymbol{x})v_t\left(\boldsymbol{x}\right)\right) d\boldsymbol{x} \right| = \tag{66}$$

$$\left| \int \left\langle \nabla\delta(\boldsymbol{x} - \gamma_t^i), -\frac{d}{dt}\gamma_t^i \right\rangle + \left\langle \nabla\delta(\boldsymbol{x} - \gamma_t^i), u(\boldsymbol{x}) \right\rangle + \delta(\boldsymbol{x} - \gamma_t^i)\operatorname{div}(u(\boldsymbol{x})) d\boldsymbol{x} \right| = \tag{67}$$

$$\left| \int \left\langle \nabla\delta(\boldsymbol{x} - \gamma_t^i), u(\boldsymbol{x}) - \frac{d}{dt}\gamma_t^i \right\rangle d\boldsymbol{x} + \int \delta(\boldsymbol{x} - \gamma_t^i)\operatorname{div}(u(\boldsymbol{x})) d\boldsymbol{x} \right|. \tag{68}$$

Now, under the assumption that $\operatorname{div}(u) = 0$ almost everywhere, using 62 yields that the second term in the last equation vanishes. Therefore,

$$0 = \left| \int \left\langle \nabla\delta(\boldsymbol{x} - \gamma_t^i), u(\boldsymbol{x}) - \frac{d}{dt}\gamma_t^i \right\rangle d\boldsymbol{x} \right| = \tag{69}$$

$$\left| \int \delta(\boldsymbol{x} - \gamma_t^i)\left\langle \nabla\log\delta(\boldsymbol{x} - \gamma_t^i), u(\boldsymbol{x}) - \frac{d}{dt}\gamma_t^i \right\rangle d\boldsymbol{x} \right| \geq \tag{70}$$

$$\int \delta(\boldsymbol{x} - \gamma_t^i)\left\| \nabla\log\delta(\boldsymbol{x} - \gamma_t^i) \right\| \left\| u(\boldsymbol{x}) - \frac{d}{dt}\gamma_t^i \right\| d\boldsymbol{x}, \tag{71}$$

where we applied the Cauchy-Schwarz inequality in the final step. Therefore,

$$\int \delta(\boldsymbol{x} - \gamma_t^i)\left\| \nabla\log\delta(\boldsymbol{x} - \gamma_t^i) \right\| \left\| u(\boldsymbol{x}) - \frac{d}{dt}\gamma_t^i \right\| d\boldsymbol{x} = 0. \tag{72}$$

Applying property 62 yields that equation 72 can be true only if when $\boldsymbol{x} = \gamma_t^i$, we have that,

$$\left\| u(\boldsymbol{x}) - \frac{d}{dt}\gamma_t^i \right\| = 0. \tag{73}$$

Utilizing this constraint for each $i$, we can derive equation 9.

## 8.2 Additional Implementation Details

### 8.2.1 Architecture

We first describe the construction of the Gaussian Splatting dynamic image model referenced in section 5. An illustration of this model is presented in Figure 5. Figure 6 illustrates the applied losses. The time invariant base of the model is optimized throughout training and consists of the following set of parameters $\boldsymbol{\theta} = \left\{ \boldsymbol{\mu}^i, \boldsymbol{S}^i, \boldsymbol{R}^i, \boldsymbol{c}^i, \alpha^i \right\}_{i=1}^{n}$ with Gaussian mean $\boldsymbol{\mu}^i \in \mathbb{R}^3$, scaling $\boldsymbol{S}^i \in \mathbb{R}^3$, rotation quaternion $\boldsymbol{R}^i \in \mathbb{R}^4$, color $\boldsymbol{c}^i \in \mathbb{R}^3$ and opacity $\alpha^i \in \mathbb{R}$. The covariance matrix $\boldsymbol{\Sigma}^i$ is calculated during the rendering process from the temporally augmented scaling and rotation parameters.

The time dependent deformation model transforms the time invariant Gaussian mean $\boldsymbol{\mu}^i$ and selected time $t$ into the deformation of the mean, scaling, rotation and the model element $w$ in the case of the adaptive-combination prior.

We generate positional embeddings (Mildenhall et al., 2020) of the time and mean inputs, which we pass to the deformation model Multilayer perceptrons.

$$\operatorname{Emb}_{\text{time}}(t) : \mathbb{R} \rightarrow \mathbb{R}^{d_{\text{time emb}}}$$

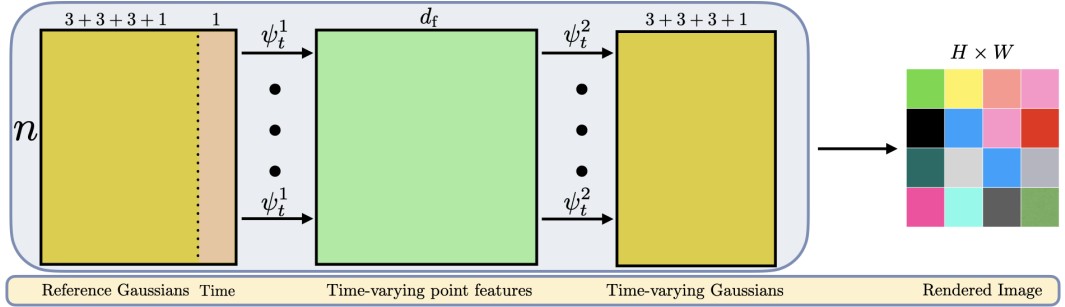

Figure 5: Illustration of the architecture for $\Psi_t$ used in the experiments, based on (Yang et al., 2023). Reference Gaussians parameters are propagated to time $t$ through a shared function, $\psi_t^1$, implemented as an MLP with positional encoding features to compute time-varying point features of dimension $d_f$. These features are then processed by a second shared function, $\psi_t^2$, to generate time-varying Gaussians parameters. Finally, given a chosen viewing direction, the Gaussian Splatting rendering model is used to produce a rendered image.

$$L_{\mathrm{Rec}}\left( \begin{matrix} \end{matrix} , \begin{matrix} \end{matrix} \right) \bigg| L_{\mathrm{RM}}\left( \begin{matrix} \end{matrix} \right) \quad L_{\mathrm{RM}}\left( \boxed{n \quad 3} \right)$$

Rendered Image   Input Image      Rendered Image   Time-varying Gaussians positions

Figure 6: Illustration of the losses applied for learning $\Psi_t$. The reconstruction loss, $L_{\mathrm{Rec}}$ is applied on a rendered image to recover a corresponding given image of the scene. The ReMatching loss, $L_{\mathrm{RM}}$, can be applied either on a rendered image or time-varying Gaussians positions.

$$\mathrm{Emb}_{\mathrm{mean}}(\mu^i) : \mathbb{R}^3 \to \mathbb{R}^{d_{\mathrm{mean\,emb}}}$$

The deformation model is made up of layers of the form:

$$\psi(n, d_{\mathrm{in}}, d_{\mathrm{out}}) : \boldsymbol{X} \mapsto \nu\left(\boldsymbol{X}\boldsymbol{W} + \mathbf{1}\boldsymbol{b}^T\right)$$

where $\nu = \mathrm{Softplus}_\beta$, with $\beta = 100$.

For the deformation of the mean, scaling and rotation, the model takes the same form with minor differences in the final layer depending on the deforming parameter.

$$\mathrm{Emb}_{\mathrm{time}}(t) \to \psi(n, d_{\mathrm{time\,emb}}, 256) \to \psi(n, 256, d_\tau) \to \tau$$

$$[\tau, \mathrm{Emb}_{\mathrm{mean}}(\mu)] \to \psi(n, d_\tau + d_{\mathrm{mean\,emb}}, 256) \to \psi(n, 256, 256) \to$$
$$\psi(n, 256, 256) \to \psi(n, 256, 256) \to [\tau, \mathrm{Emb}_{\mathrm{mean}}(\mu), \psi(n, 256, 256)] \to$$
$$\psi(n, d_\tau + d_{\mathrm{mean\,emb}} + 256, 256) \to \psi(n, 256, 256)] \to \psi(n, 256, 256)] \to \omega$$

$$\mathrm{Mean} : \omega \to \psi(n, 256, 3) \to \mu(t)$$
$$\mathrm{Scaling} : \omega \to \psi(n, 256, 3) \to \boldsymbol{S}(t)$$
$$\mathrm{Rotation} : \omega \to \psi(n, 256, 4) \to \boldsymbol{R}(t)$$

For the prediction of the $w$ we use a shallower Multilayer perceptron.

$$[\tau, \mathrm{Emb}_{\mathrm{mean}}(\mu + \mu(t)), \mathrm{Emb}_{\mathrm{mean}}(\mu)] \to \psi(n, d_\tau + 2 \cdot d_{\mathrm{mean\,emb}}, 256) \to$$
$$\psi(n, 256, K) \to \mathrm{Softmax} \to w(t)$$

### 8.2.2 HYPER-PARAMETERS AND TRAINING DETAILS

We set $d_{\text{mean emb}} = 63$, $d_{\text{time emb}} = 13$ and $d_\tau = 30$. For optimization we use an Adam optimizer with different learning rates for the network components, keeping the hyper-parameters of the baseline model (Yang et al., 2023).

In the case of the adaptive-combination prior we select $k$ based on a hyper-parameter search between 1 and 35. The optimal value for most scenes ranges between 5 and 15, though the number also depends on the selected composition of priors. For example, a single volume-preserving class can supervise multiple moving objects as opposed to a single rigid deformation class. We use the ReMatching loss weight $\lambda = 0.001$. When supplementing the ReMatching loss with an additional entropy loss, we use 0.0001 as the entropy loss weight.

In calculating the partial derivatives of $\psi_t$ , we note that the input dimension for predicting $\psi_t$ is relatively small − 1 for time or $d$ for spatial coordinates − compared to the output dimension, which can be $n \times d$ for spatial ReMatching or $H \times W$ in image-space ReMatching. Given this, forward-mode automatic differentiation proves to be more efficient than backward-mode differentiation for this specific computation, both computationally and in terms of memory usage. Consequently, we utilize forward-mode autodiff to compute the partial derivatives of $\psi_t$ required for the ReMatching loss. Once the ReMatching loss is incorporated, we employ backward-mode autodiff to compute the gradient of the overall loss with respect to the model parameters.

### 8.2.3 REMATCHING RENDERED IMAGE

The reconstruction model architecture in the case of the image ReMatching is the same as for the other experiments. At initialization we select a fixed viewpoint for the evaluation of the image space loss, which is kept throughout training. At every iteration we sample a random time and evaluate the ReMatching loss from the fixed viewpoint.

For approximating equation 10, we calculate a sample by choosing points that their image value is close to 0 after applying the following transformation:

$$f(x) = -0.1 \cdot \ln(1 - |x|) \cdot \text{sign}(x) \tag{74}$$

on the image.

Next, we compute the image gradient using automatic differentiation and use our single class div-free solver to reconstruct the flow and calculate the loss.

### 8.3 ADDITIONAL EVALUATION

### 8.3.1 3D GEOMETRY-AWARE DEFORMABLE GAUSSIANS

We provide an additional set of experiments that show the versatility of our framework. Our evaluations of the framework in tables 1 and 2 were made by applying our framework to a reconstruction model inspired by Deformable 3D Gaussians (Yang et al., 2023). In this evaluation, we apply the same framework on a baseline following the 3D Geometry-aware Deformable Gaussians (Lu et al., 2024). Table 3 shows our results compared to the baseline GA3D.

### 8.3.2 DYNAMIC SCENES DATASET

To further evaluate the efficacy of the ReMatching framework in practical applications, we consider the Dynamic Scenes dataset (Yoon et al., 2020), which captures forward-facing views of real-world scenes exhibiting complex dynamics. To that end, we selected 4 scenes from the Human, Interaction, and Vehicle categories, consisting of monocular videos with approximately 80–180 frames for training and an additional 20 frames reserved for testing. Figure 8 shows qualitative comparison to the D3G (Yang et al., 2023) baseline, highlighting similar patterns of improvement as observed in earlier experiments. Specifically, our approach better preserves fine details, such as the truck's front lights (Truck) and the bottom teeth (Dynamic Face). Additionally, it demonstrates a reduction in reconstruction motion artifacts, as evident in the humans in motion (Jumping) and the legs and head of the dinosaur (Balloon). Table 4 presents a quantitative evaluation, comparing two variants from

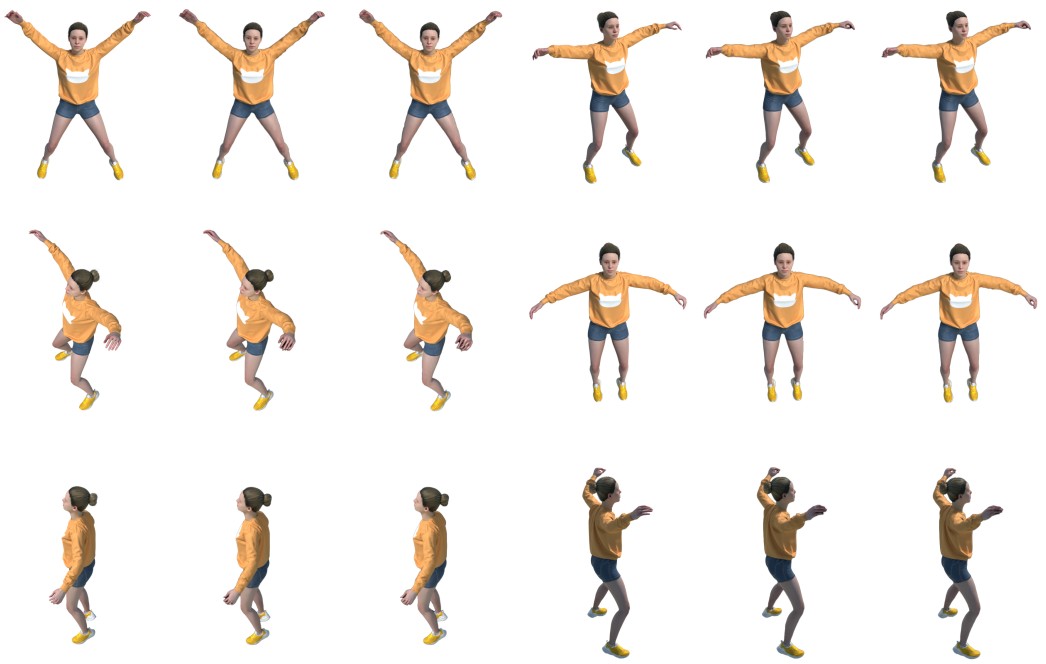

Figure 7: Qualitative comparison of the ReMatching loss applied in the image space. Each group of 3 is showing Ground-Truth (left), Ours (center), and D3G (right).

our prior classes, $\mathcal{P}_{III}$ and $\mathcal{P}_{IV}$ to D3G. These results correlate with the qualitative improvements discussed above.

## 8.4 RECONSTRUCTION FLOW EVALUATION

In this section, we evaluate the ability of the ReMatching framework to recover the underlying reconstruction flow $\phi_t$. Since the reconstruction flow is generally unknown, we use the following simple flow to generate training data:

$$\phi_t(\boldsymbol{x}) = \boldsymbol{R}(t)\boldsymbol{x} \tag{75}$$

where $\boldsymbol{R}^T(t)\boldsymbol{R}(t) = I_d$. We evaluate our framework in its two settings: i) equation 9, corresponding to $V = \mathbb{R}^{n \times d}$; and, ii) equation 10, corresponding to $V = C^1(\mathbb{R}^d)$.

**The $V = \mathbb{R}^{n \times d}$ case.** We evaluate these settings for $d = 3$, following a similar approach to the dynamic image model based on Gaussian Splatting described in Section 5. To construct the training set, we use a reference scene of a colored box and apply the ground-truth flow $\phi_t$, to create a dynamic scene consisting of multi-view captures over 12 time stamps. For the flow $\phi_t$, we set

| | **Bouncing Balls** | | | **Hell Warrior** | | | **Hook** | | | **JumpingJacks** | | |
|---|---|---|---|---|---|---|---|---|---|---|---|---|
| Method | LPIPS ↓ | PSNR ↑ | SSIM ↑ | LPIPS ↓ | PSNR ↑ | SSIM ↑ | LPIPS ↓ | PSNR ↑ | SSIM ↑ | LPIPS ↓ | PSNR ↑ | SSIM ↑ |
| GA3D (Lu et al., 2024) | 0.0063 | 43.42 | 0.9960 | 0.0175 | 32.02 | 0.9827 | 0.0076 | 36.88 | 0.9915 | 0.0092 | 37.83 | 0.9924 |
| Ours ($\mathcal{P}_{III}$) | 0.0061 | 43.58 | 0.9961 | 0.0165 | 32.15 | 0.9834 | 0.0075 | 36.92 | 0.9914 | 0.0093 | 37.84 | 0.9927 |
| Ours ($\mathcal{P}_{IV}$ or $\mathcal{P}_V$) | 0.0058 | 43.62 | 0.9962 | 0.0163 | 32.21 | 0.9835 | 0.0077 | 37.01 | 0.9916 | 0.0078 | 38.18 | 0.9931 |
| | **Lego** | | | **Mutant** | | | **Stand Up** | | | **T-Rex** | | |
| Method | LPIPS ↓ | PSNR ↑ | SSIM ↑ | LPIPS ↓ | PSNR ↑ | SSIM ↑ | LPIPS ↓ | PSNR ↑ | SSIM ↑ | LPIPS ↓ | PSNR ↑ | SSIM ↑ |
| GA3D (Lu et al., 2024) | 0.0328 | 25.41 | 0.9471 | 0.0029 | 41.56 | 0.9969 | 0.0028 | 42.40 | 0.9967 | 0.0055 | 38.65 | 0.9950 |
| Ours ($\mathcal{P}_{III}$) | 0.0316 | 25.53 | 0.9489 | 0.0029 | 41.41 | 0.9968 | 0.0028 | 42.24 | 0.9967 | 0.0052 | 39.12 | 0.9952 |
| Ours ($\mathcal{P}_{IV}$ or $\mathcal{P}_V$) | 0.0320 | 25.50 | 0.9487 | 0.0028 | 41.65 | 0.9970 | 0.0027 | 42.44 | 0.9968 | 0.0049 | 39.18 | 0.9953 |

Table 3: Image quality evaluation on unseen frames for the D-NeRF dataset with our framework applied on top of the 3D Geometry-aware Deformable Gaussians (Lu et al., 2024). Image resolution is scaled down to 400x400 pixels and the background is white, maintaining the settings of the GA3D paper evaluation.

| Method | Balloon | | | Truck | | | Jumping | | | Dynamic Face | | |
|---|---|---|---|---|---|---|---|---|---|---|---|---|
| | LPIPS ↓ | PSNR ↑ | SSIM ↑ | LPIPS ↓ | PSNR ↑ | SSIM ↑ | LPIPS ↓ | PSNR ↑ | SSIM ↑ | LPIPS ↓ | PSNR ↑ | SSIM ↑ |
| D3G (Yang et al., 2023) | 0.1584 | 26.79 | 0.9349 | 0.2922 | 26.01 | 0.9046 | 0.2726 | 23.12 | 0.8958 | 0.0806 | 29.22 | 0.9756 |
| Ours - $\mathcal{P}_{III}$ | 0.1592 | 26.96 | 0.9348 | 0.2782 | 25.49 | 0.9071 | 0.2720 | 22.89 | 0.8971 | 0.0794 | 29.30 | 0.9761 |
| Ours - $\mathcal{P}_{IV}$ | 0.1578 | 26.95 | 0.9356 | 0.2533 | 26.66 | 0.9197 | 0.2501 | 23.63 | 0.9037 | 0.0793 | 29.23 | 0.9754 |

Table 4: Unseen frames evaluation for the dynamic scenes dataset (Yang et al., 2023).

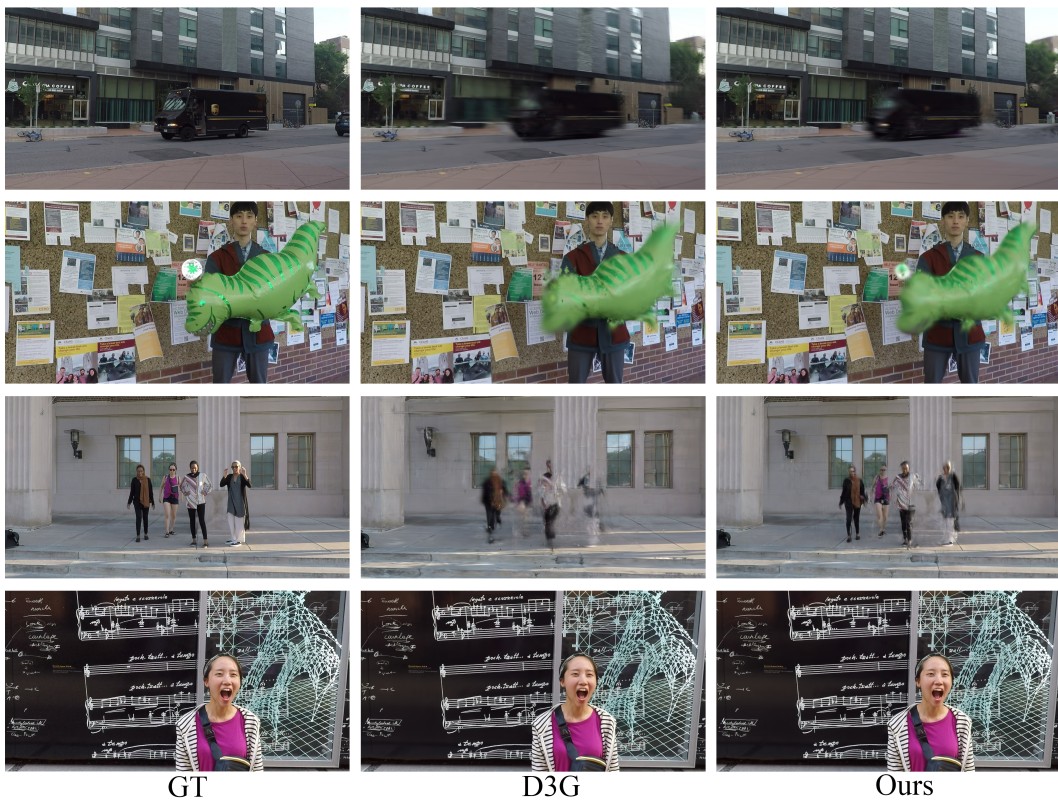

|  GT  |  D3G  |  Ours  |

Figure 8: Qualitative comparison of our method to D3G (Yang et al., 2023) on the dynamic scenes dataset (Yoon et al., 2020).

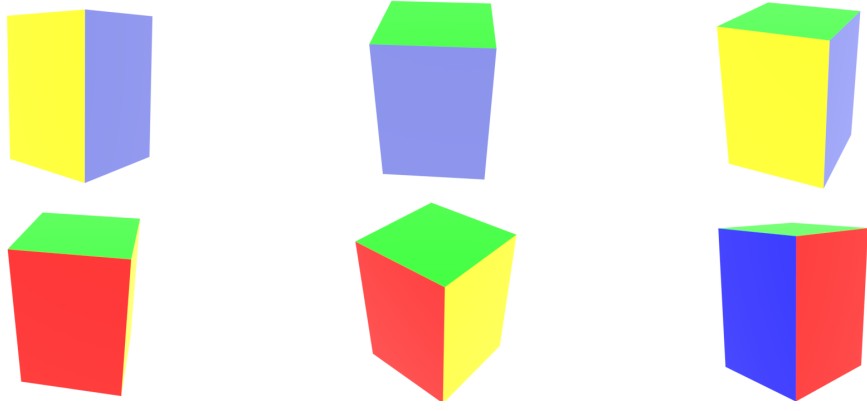

Figure 9: Training frames from the rotating colored box scene.

$$\boldsymbol{R}(t) = \begin{bmatrix} \cos 2\pi t & -\sin 2\pi t & 0 \\ \sin 2\pi t & \cos 2\pi t & 0 \\ 0 & 0 & 1 \end{bmatrix}.$$ Figure 9 shows selected images from the training set. Two training procedures are considered: (i) a baseline approach using only the reconstruction loss, similar to the

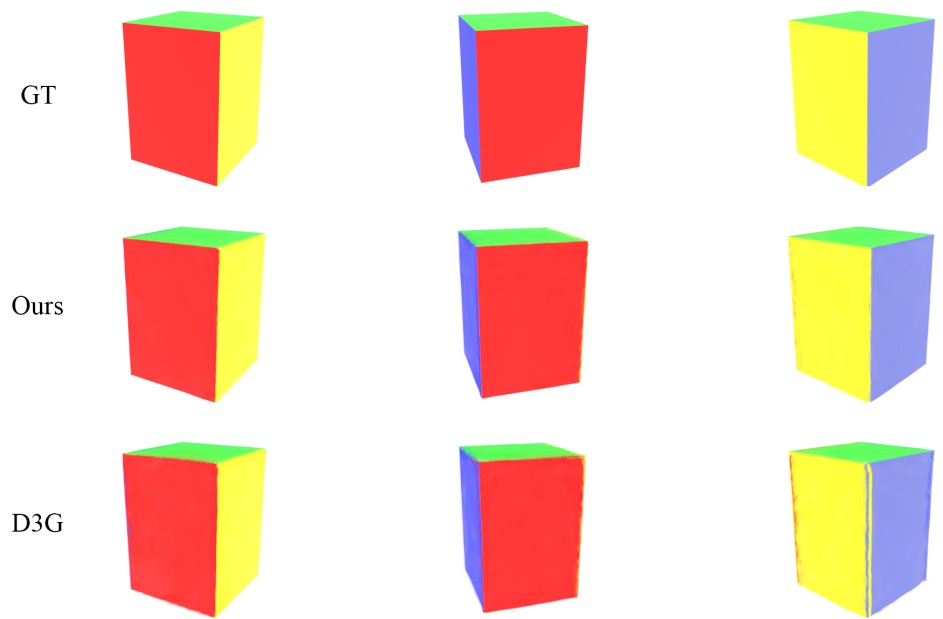

Figure 10: Evaluation of unseen timestamps in the rotating colored box scene.

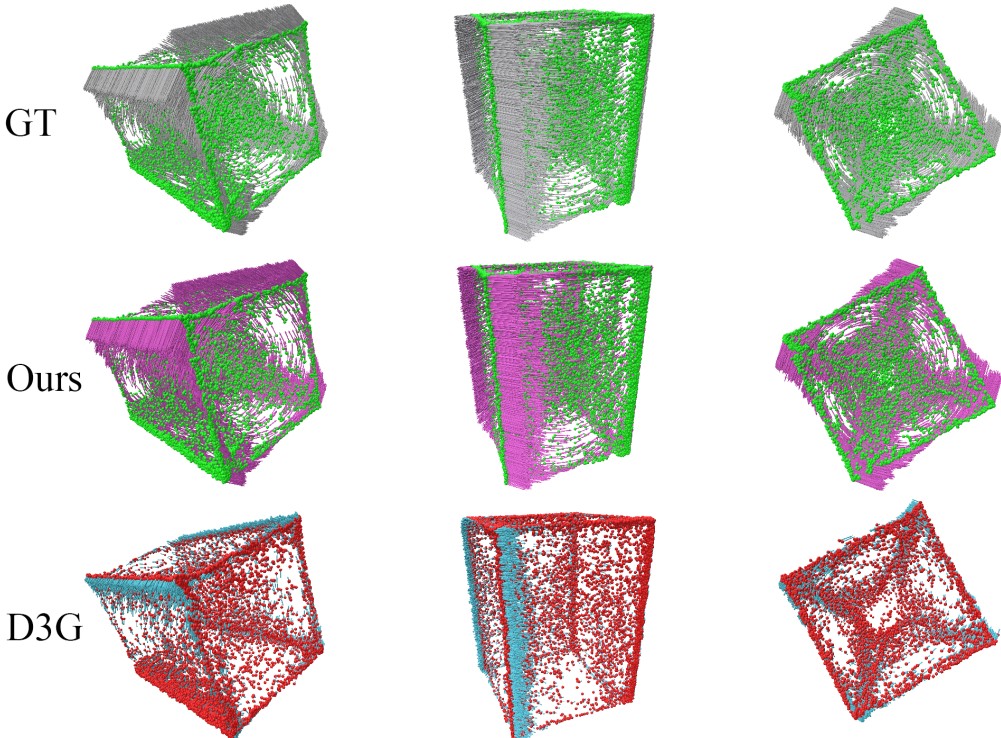

Figure 11: Visualization of $\{\mu^i(t), \dot{\mu}^i(t)\}$ from multiple timestamps.

D3G model; and, (ii) our approach where both the reconstruction loss and the ReMatching loss are optimized. For the ReMatching loss, we employ the global rigid motion prior class $\mathcal{P}_{II}$. Figure 10 compares ground-truth images from time-stamps unseen during training with the model's predicted renderings. Notably, the ReMatching loss allows the model to generalize in alignment with the ground-truth flow $\phi_t$. This is a result of the ReMatching objective's ability to converge to matched priors $u_t$ that accurately recover the ground truth velocities $\frac{\partial}{\partial t}\phi_t$. To further support this claim, Figure

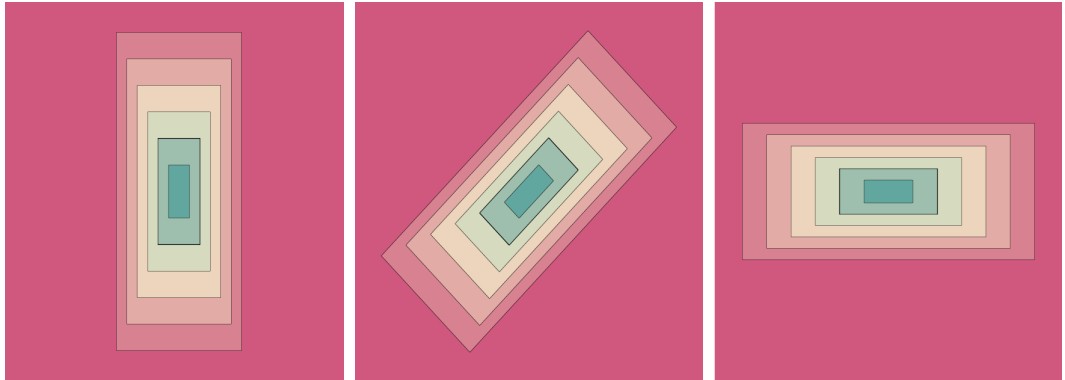

Figure 12: Visualization of the training set made up of three functions $\psi_{\text{GT}}(\cdot, t_i)$.

11 illustrates the velocities of the dynamic Gaussian centers, $\{\dot{\mu}^i(t)\}$. These results demonstrate that the ReMatching loss effectively controls $\{\dot{\mu}^i(t)\}$, resulting with solutions $\{\dot{\mu}^i(t)\}$ that match the prior class $\mathcal{P}_{II}$ and recover the ground-truth velocities $\frac{\partial}{\partial t}\phi_t$.

**The $V = C^1(\mathbb{R}^d)$ case.** We evaluate these settings for $d = 2$. Using the flow described above, we define the following ground-truth scalar function, $\psi_{\text{GT}} : \mathbb{R}^2 \to \mathbb{R}$, as:

$$\psi_{\text{GT}}(\boldsymbol{x}, t) = \left\|\phi_t^{-1}(\boldsymbol{S}^{-1}\boldsymbol{x})\right\|_\infty - b. \tag{76}$$

The training data is constructed using three time-stamps, specifically $t \in \{0.0, 0.25, 0.5\}$. The parameter choices for this procedure are: $b = 0.2$, and $\boldsymbol{S} = \text{diag}\{0.6, 1.4\}$, $\boldsymbol{R}(t) = \begin{bmatrix} \cos\pi t & \sin\pi t \\ -\sin\pi t & \cos\pi t \end{bmatrix}$. Figure 12 visualizes the three distinct functions $\psi_{\text{GT}}(\cdot, t)$ that constitute the training set. To model $\psi_t$, we employ a multi-layer perceptron (MLP) architecture, as described in Section 8.2.1, with the only modification of a scalar output dimension in the final layer. For the reconstruction loss, we adopt the standard $L_1$ loss:

$$L_{\text{REC}} = \sum_{i=1}^{3} \mathbb{E} \|\psi(\boldsymbol{x}, t_i) - \psi_{\text{GT}}(\boldsymbol{x}, t_i)\|. \tag{77}$$

For the ReMatching loss, we employ the global rigid motion prior class $\mathcal{P}_{II}$. Two training procedures are considered: (1) a baseline approach where only the reconstruction loss is used as the optimization objective, and (2) our suggested approach where both the reconstruction loss and the ReMatching loss are optimized. Figure 13 displays the results of the trained models for times $t \in \{0, 0.0625, 0.125, 0.1875, 0.25, 0.3125, 0.375, 0.4375\}$. Among these, only $t \in \{0, 0.25, 0.5\}$, shown in the leftmost column, correspond to the training set frames. While both the baseline and our approach perform similarly on the training frames, the results for unseen frames clearly demonstrate the benefits of incorporating the ReMatching loss. Specifically, the ReMatching loss allows the model to recover the ground-truth flow $\phi_t$, avoiding the unrealistic distortions observed in the baseline results. To further illustrate this, Figure 14 depicts the matched priors $u_t$ (white arrows) obtained by solving equation 5, alongside the velocity field of the ground-truth flow $\frac{\partial}{\partial t}\phi_t$ (green arrows). These comparisons show that the ReMatching loss successfully converges to matched priors $u_t$ that closely approximate the ground truth. In contrast, the matched priors $u_t$ obtained with the baseline approach (without the ReMatching loss) deviate significantly from the ground truth. This emphasizes the importance of the reprojection procedure. Not all velocity fields in the prior class are suitable for guiding the optimization process, but the ReMatching loss ensures convergence to an appropriate prior, enabling an accurate recovery of the underlying flow.

## 8.5 RUNTIME AND CONVERGENCE ANLAYSIS

We note that our framework is applied solely during the training phase of the algorithm, leaving inference times unaffected. To evaluate computational efficiency, we measured the average time (seconds) for a forward and backward pass over 100 iterations for varying sizes $n$ of Gaussians sets.

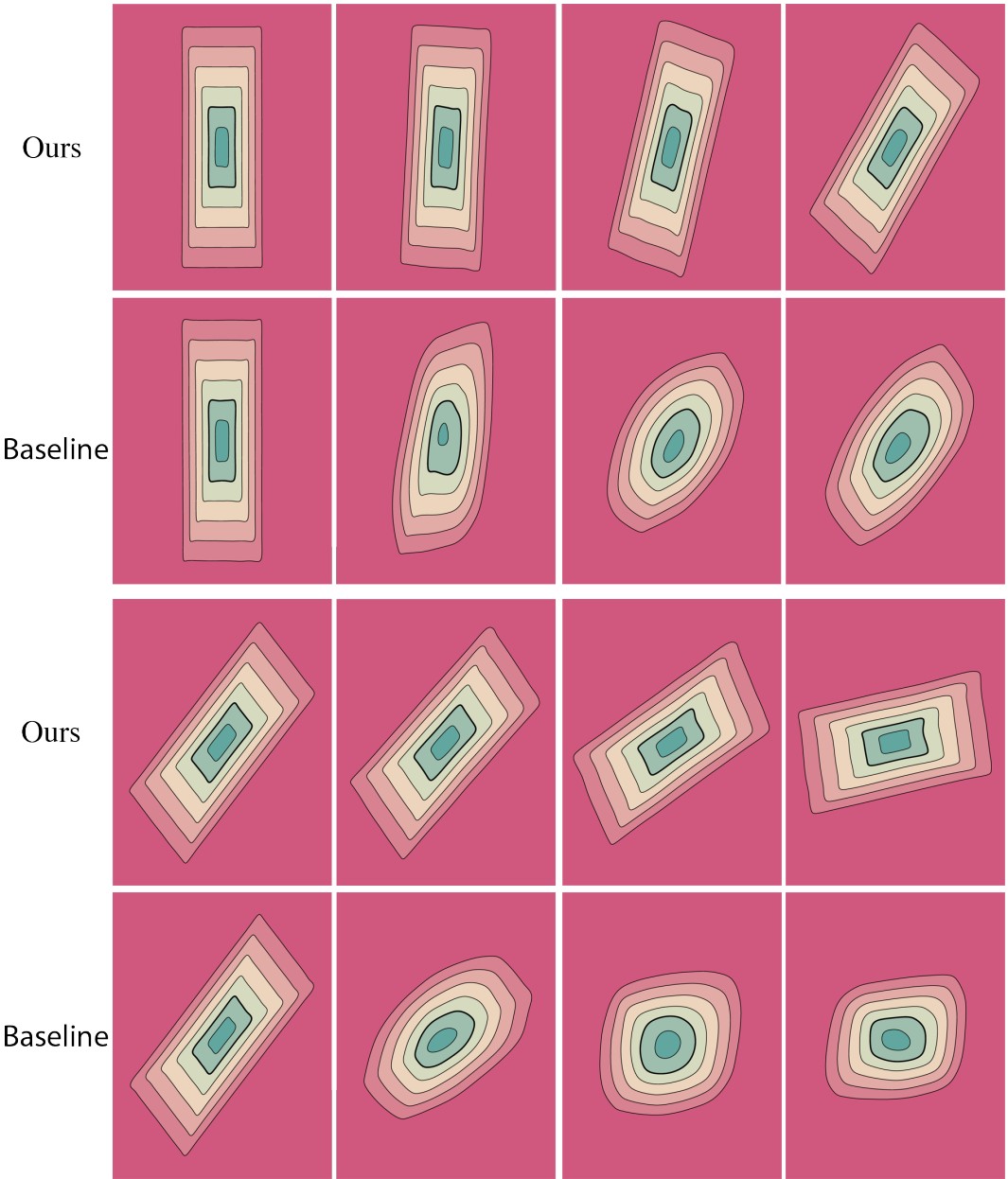

Figure 13: Comparisons of $\psi_t$ converged solutions between the baseline and our approach, displayed in order of increasing time (left to right, top to bottom).

Figure 15 presents the results, comparing the computation time of the ReMatching framework to the D3G baseline. The runtime analysis was conducted on a single NVIDIA RTX A6000.

To examine the convergence of the reconstruction model, we compare the loss convergence curves of the D3G (Yang et al., 2023) model and our model. Figure 16 shows that the addition of the ReMatching loss does not affect the convergence behavior of the optimization. We also show the loss curve of theReMatching loss itself in figure 17. It is important to note that for the ReMatching formulation, the optimal solution does not necessarily achieve 0 loss, simliarly to the APM procedure (Deutsch, 1992). Instead, it achieves the lowest loss possible given the reconstruction task and selected prior.

## 8.6 ABLATION OF FRAMEWORK HYPERPARAMETERS

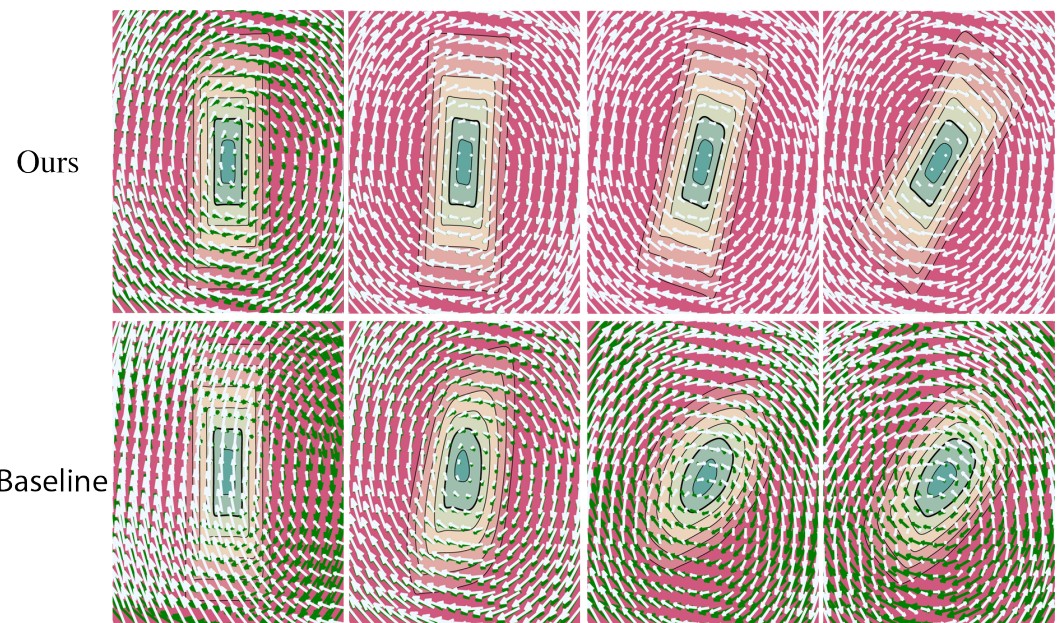

Figure 14: Comparisons of the converged $u_t$ (white arrows) between the baseline and our approach that uses the ReMatching loss. The ground-truth velocities, $\frac{\partial}{\partial t}\phi_t$, are shown as green arrows. When the matched $u_t$ aligns with the ground truth, the green arrows become indistinguishable.

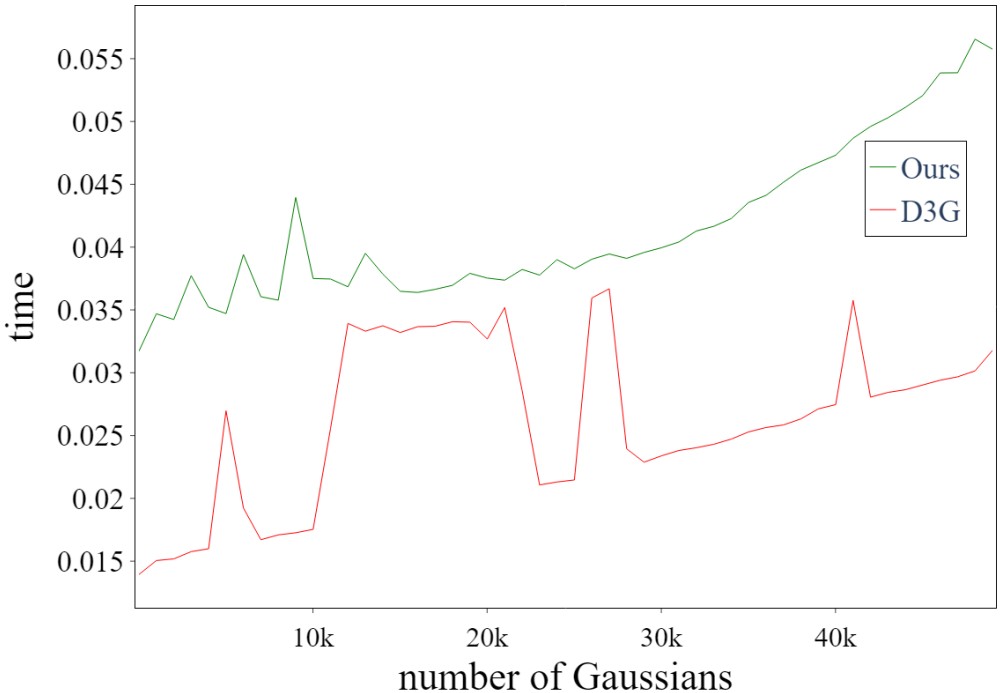

Figure 15: Combined average time (seconds) for a forward and backward pass for varying size of $n$.

In this section, we present an ablation study on key hyperparameters introduced by the ReMatching framework: i) the weight of the ReMatching loss, $\lambda$, as defined in equation 8; ii) maximum number of parts selection, $k$, for the adaptive prior class; and iii) the weight of the entropy loss (equation 29), used to optimize the learned part assignments when employing an adaptive prior class.

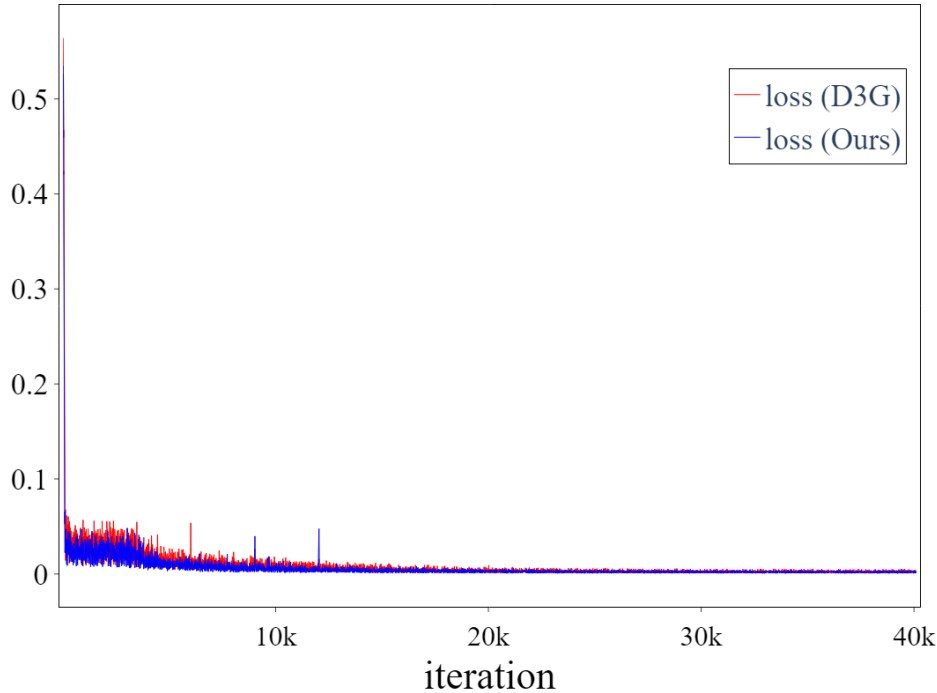

Figure 16: Loss curves report of our model and D3G (Yang et al., 2023) over 40k training iterations.

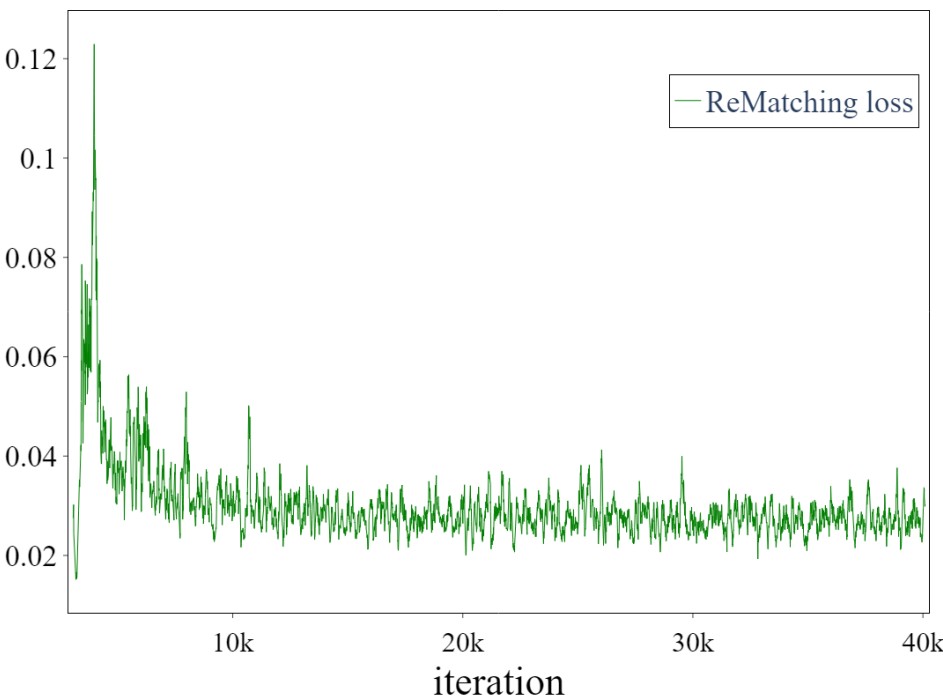

Figure 17: Loss curve report for the ReMatching loss, showing a running average with a window size of 20.

**ReMatching loss weight.** We note first that a consistent value of $\lambda = 0.001$ was used across all scenes experimented with in section 6, already demonstrating the robustness of this parameter. To further test this, we conducted experiments on the Hell Warrior and Lego scenes from the D-NeRF

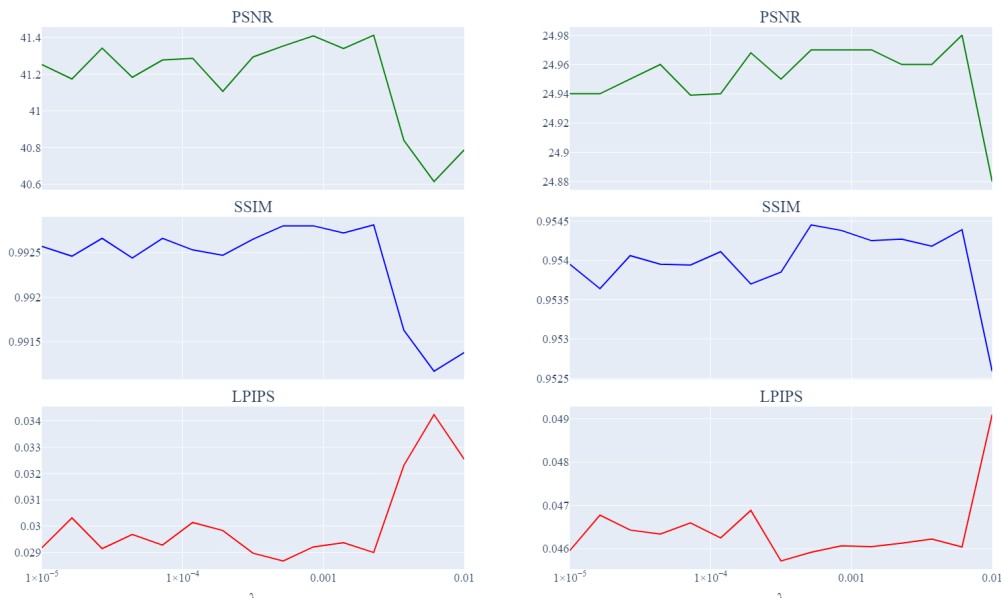

Figure 18: Effect of the weight parameter $\lambda$ on the PSNR evaluation metric for the Hell Warrior scene (left) and the Lego scene (right).

dataset, evaluating how different $\lambda$ values influence solution quality. Figure 18 shows these findings. We note that for the Hell Warrior scene, we employed the $\mathcal{P}_{IV}$ prior class, while the Lego scene used the $\mathcal{P}_V$ class. The results indicate stable improvement within the range of $\lambda \in [5e{-}4, 5e{-}3]$, while small values $\lambda \leq 5e{-}5$ aligns with the baseline. Larger values, $\lambda \geq 1e{-}2$, may compete with the reconstruction loss, leading to suboptimal solutions.

**Maximum number of parts selection.** To assess the impact of the hyperparameter $k$, we selected the Mutant scene, which aligns with the adaptive prior class $\mathcal{P}_{IV}$, and the Lego scene, corresponding to the adaptive prior class $\mathcal{P}_V$ and evaluated how varying $k$ affects solution quality. Figure 19 presents the results of this analysis. The findings suggest relatively stable performance within the range $k = 5$ to $k = 15$, offering flexibility in selecting $k$ based on leveraging prior knowledge about the expected number of moving parts in the scene. As a special case for $k = 1$, the adaptive prior $\mathcal{P}_V$ corresponds to $\mathcal{P}_I$. If we train with ReMatching using this prior on the three selected scenes we get the following quantitative results: Lego (PSNR: 24.94, SSIM: 0.9536, LPIPS: 0.0452), Hell Warrior (PSNR: 40.77, SSIM: 0.9917, LPIPS: 0.0285), Mutant (PSNR: 41.88, SSIM: 0.9965, LPIPS: 0.0069).

**Entropy loss weight.** Similar to the $\lambda$ hyperparameter, the entropy loss weight was kept fixed across all experiments in section 6. To further examine its impact, we evaluated its influence on performance with varying weight values for the Hell Warrior scene. Figure 20 presents the results of this experiment, demonstrating stable performance within the range $[1e{-}4, 1e{-}3]$.

## 8.7 ADDITIONAL QUALITATIVE EVALUATION

To further support the qualitative results presented in Figures 2 and 3, the supplementary material includes additional evidence showcasing novel-view video reconstruction results. These comparisons highlight the performance of our model relative to baseline approaches, providing a more comprehensive validation of its efficacy.

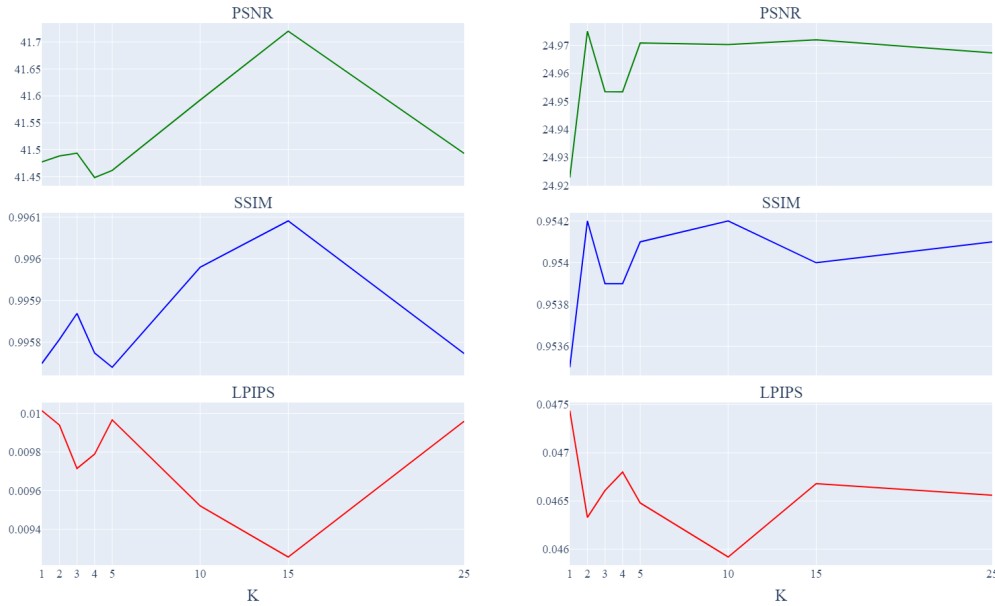

Figure 19: Impact of varying $k$ values on the PSNR evaluation metric for the Mutant scene (left) and the Lego scene (right).

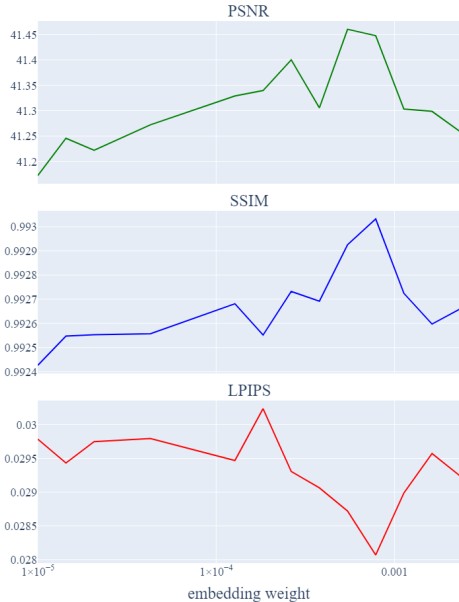

Figure 20: Impact of varying entropy loss weights on the PSNR evaluation metric for the Hell Warrior scene.

