# OpenReview forum: "ReMatching Dynamic Reconstruction Flow"
_ICLR.cc/2025/Conference — ICLR 2025 Poster_

### Official Review · Reviewer_SSgx · 2024-10-30

**Soundness:** 3
**Presentation:** 3
**Contribution:** 3
**Rating:** 8
**Confidence:** 4

**Summary:**

This paper tackles the problem of dynamic view synthesis (with multi-view input). To model the time-dependent function, it introduces a flow-matching mechanism. Essentially, by assuming that a flow or velocity field exists that can reconstruct the true time-dependent functions with some predefined flow priors, e.g., rigid or volume-preserving, this paper utilizes the alternating projection method (AMP) to guide the training of the velocity field. Further, with some specific time-dependent function format, the authors demonstrate that there exist closed-form solutions to such AMP problems. Empirical experiments on D-NeRF and HyperNeRF datasets demonstrate the effectiveness of the proposed approach.

**Strengths:**

- originality-wise: I appreciate the principled way that the authors formulate the problem of dynamic view synthesis as a prior-guided flow-matching problem. This will be useful and interesting to the community.
- quality-wise: the approach presented in the paper is solid and the experiments demonstrate the effectiveness.
- clarify-wise: overall, the paper is easy to follow.
- significance: the task of dynamic view synthesis is of great importance to many downstream applications, e.g., AR/VR.

**Weaknesses:**

## Confusion about Eq. (9)

I am quite confused with the definition used to derive Eq. (9). Concretely, the authors state $V = \mathbb{R}^n$ in L202, which is an Euclidean space. However, the authors further define $\psi_t  = (\gamma_t^1, \dots, \gamma_t^n)^T$ where $\gamma^i: \mathbb{R}_+ \mapsto \mathbb{R}^d$ is a function. Do these conflict?

Further, I got confused about how Eq. (9) is derived based on the above definition. I tried to follow the product rule for the divergence, i.e., L212. However, I am unable to reach the result, especially $u_t (\gamma_t^i)$. Can authors provide details?

## Inadequate visualization

Since the underlying mechanism is to train a velocity field $v$ (Eq. (2)) and its corresponding flow $\phi_t$ (Eq. (3)), I am quite curious to know 1) how the reconstructed flow as well as velocity fields look like; 2) how the prior set $\mathcal{P}$ (L153) look like; and 3) the difference between the prior and the trained flow.

I think this is important such that readers can be well aware of the performance of the approach.

## Speed

Can the authors clarify the efficiency of the flow-matching approach? I think the main overhead compared to vanilla Gaussian Splats comes from the computation of the ReMatching loss $L_\text{RM}$ (Eq. (7)).

If I understand correctly, in Algorithm 1, the $\texttt{solve}$ step in L189 will be solved analytically based on those priors specified Sec. 4, which should be quick. Further, the computation of $\rho$ in L190 should also be quick due to those priors. Can authors clarify whether this is the case?

## Missed reference

The following work utilizes an ODE solver to conduct dynamic reconstruction from videos, which is quite relevant to this work.

[a] Class-agnostic Reconstruction of Dynamic Objects from Videos. NeurIPS 2021.

**Questions:**

1. Can authors clarify how to make HyperNeRF a multi-view dynamic view synthesis benchmark? Essentially, HyperNeRF only has two views. If the authors use up those two views, how to evaluate view synthesis results? Or do we only conduct time-interpolation evaluations? If the authors indeed carry out monocular dynamic view synthesis, please correct L100 as the "multi-view images" statement is not correct.

2. In L131, the authors state that there is a reference function $\psi_0$ to be pushed forward. Concretely, does $\psi_0$ refer to the initialized Gaussian mean $\mu^i$ in Eq. (29)?

3. In L849, the authors state that they use **forward-mode** auto differentiation. Can authors clarify the reason of not using the traditional backpropagation, i.e., reverse-mode? Essentially, what prevents from using that?

---

> ### Author Response · Authors · 2024-11-20
>
> We thank the reviewer for a detailed review. Below we address the main comments and questions expressed in this review.
>
> **W1:** “I am quite confused with the definition used to derive Eq. (9)”.
>
> **A:** Thank you for taking the time to spot this. Indeed it should have been $V=\mathbb{R}^{n\times d}$. We have corrected this typo in the latest revision.
>
> **W2:** “how Eq. (9) is derived based on the above definition. Can authors provide details?”.
>
> **A:** Thank you for this question. We agree with the reviewer on the value of clarifying this in the text. As a result, we added these details to section 8.1.2 in the appendix, included in the latest revision. The formal way to utilize discrete samples of trajectories ${\gamma_t^i}$ within the continuity equation (4) is by considering $\psi_t(x) = \sum_{i=1}^n \delta(x - \gamma_t^i)$, where $\delta(\cdot - a)$ is the dirac-delta generalized function concentrated around $a$. An alternative approach is to consider $\gamma_t^i$ as samples of integral curves of the (unknown) reconstruction flow, and utilize equation (3) to derive equation (9). However, due to the generality of the continuity equation (covering the discrete settings as a specific case), in the paper we followed the former approach suggesting utilizing equation (4).
>
> **W3:** Inadequate visualization.”I am quite curious to know 1) how the reconstructed flow as well as velocity fields look like; 2) how the prior set look like; and 3) the difference between the prior and the trained flow.”
>
> **A:** Thank you for this insightful question. We agree with the reviewer on the importance of this visualization. We will include it in the next revision, to be uploaded before the end of the discussion period.
>
> **W4:** Can the authors clarify the efficiency of the flow-matching approach? “If I understand correctly, in Algorithm 1, the  step in L189 will be solved analytically based on those priors specified Sec. 4, which should be quick. Further, the computation of  in L190 should also be quick due to those priors”.
>
> **A:** The reviewer’s description is accurate. To support this analysis empirically, we have included timing comparisons of the ReMatching loss in the appendix, included in the latest revision.
>
> **W5:** Missed reference
>
> **A:** Thank you for spotting this. We have added this reference to the latest revision.
>
> **Q1:** Can authors clarify how to make HyperNeRF a multi-view dynamic view synthesis benchmark? If the authors indeed carry out monocular dynamic view synthesis, please correct L100 as the "multi-view images" statement is not correct.
>
> **A:** Thank you for spotting this. We have evaluated HyperNeRF under monocular dynamic view synthesis settings. We clarified the settings of which our method operates in section 3, now included in the latest revision.
>
> **Q2:** The authors state that there is a reference function to be pushed forward. Concretely, does $\psi_0$ refer to the initialized Gaussian mean $\mu^i$ in Eq. (29)?
>
> **A:** Yes, in this instantiation, the $\mu^i$ serve as the references to be pushed forward. As a side remark, we note that one advantage of the matching formulation is its independence from the reference function $\psi_0$. Instead, it relies solely on the derivatives at intermediate times, allowing control over the flow deformation without the necessity of pushing forward the reference function.
>
> **Q3:** The authors state that they use forward-mode auto differentiation. Can authors clarify the reason of not using the traditional backpropagation, i.e., reverse-mode?
>
> **A:** To clarify, we have utilized forward-mode autodiff only to compute the partial derivatives of
> $\psi_t$, which are necessary for calculating the ReMatching loss. Once the ReMatching loss is evaluated and incorporated into the total loss, the gradient of the total loss with respect to the model parameters is computed using standard backward-mode autodiff. This clarification has been added to the text in the latest revision (see lines 905-912).
> The rationale for employing forward-mode autodiff for$ \psi_t$ partial derivatives is that the input dimension is relatively small—either 1 (for time) or $d$ (for spatial coordinates)—compared to the output dimension, which can be $n \times d$ or, in the case of images, $H \times W$. In such cases, forward-mode autodiff is both computationally and memory-efficient compared to backward-mode autodiff.

---

> > ### Comment · Reviewer_SSgx · 2024-11-22
> >
> > I thank the authors' time and effort in addressing my concerns.
> >
> > After carefully reading other reviews and the authors' responses as well as the newly-added content in the paper, I remain positive about the submission for its principled way of guiding the training of the dynamic view synthesis model via an iterative projection mechanism. In my opinion, the analytic solutions themselves are of interest to the community as they provide a refreshing thought on how to tackle the problem. Since the authors also state that they will release the code, I think this could potentially spur future works either from more empirical perspectives or from more theoretical angles.
> >
> > However, I also agree with other reviewers that the current version is not super clear about the contribution. I would suggest adding some specific statements about the contributions of the paper to ease readers' understanding.
> >
> > I am also looking forward to seeing visualizations of the priors as well as learned flows.

---

> ### Author Response · Authors · 2024-11-25
>
> We sincerely thank the reviewer for their prompt response and valuable suggestions.
>
> In line with the recommendations, the revised paper now includes visualizations of the learned reconstruction flow. A key challenge in generating these visualizations is that our framework does not explicitly model the reconstruction flow, nor it is accessible with existing datasets. To address this, we created a synthetic training dataset with dynamics governed by a predefined flow. These experiments validate the effectiveness of the ReMatching objective in recovering the underlying flow. In addition, it emphasizes the significance of the rematching procedure,  as illustrated in Figure 13, which shows that the matched velocity fields of solutions not trained with the rematching loss deviate significantly from the ground-truth generating velocity fields.
>
> Furthermore, we have addressed the reviewer’s feedback on improving readability by adding a summary paragraph to the introduction, highlighting our key contributions more clearly.

---

### Official Review · Reviewer_GNax · 2024-10-31

**Soundness:** 2
**Presentation:** 2
**Contribution:** 2
**Rating:** 5
**Confidence:** 5

**Summary:**

This paper aims to solve dynamic reconstruction task. It proposes a new loss called ReMatching loss, in order to inject some physical priors into the learned deformation field. They also propose five categories of physics priors, and derive the corresponding forms of the RM loss respectively. The paper builds the model based on the deformable-3d-gaussian paper, and tests the performance for three of the five categories on both synthetic and real-world datasets, showing comparable performance to baselines.

**Strengths:**

In presentation, this paper shows dedicatedly derived loss form and clear derivation for different kinds of velocity priors. For the idea, the loss proposed by this paper constrains the function shape of deformation field with some physics priors. Working similar as a PINN loss, this framework seems flexible to different deformation designs.

**Weaknesses:**

1. In synthetic experiments, type IV and type V priors are selected for each scene. They should be thoroughly tested respectively to show the applicability under different scenarios.

2. In real-world experiments, it is not clear which type of priors are used. And also, all of the priors should be tested to show the abilities.

3. The velocity branches in the model only serve as a constraint part, but not used directly. The paper should demonstrate whether the learned velocity is meaningful or not. For example, using the velocity to advect the scene.

4. The segmentation compared with SAM is not proper and confusing. Since SAM segments objects according to semantics but this paper segments according to motions, it’s not proper to punish SAM’s over-segmentation. Another question is about the bouncing ball example shown in the paper. As I know this scene includes some moving shadow on the ground, so the motion should not segment the shadow part as the same as the static ground. How is the segmentation applied?

5. The performances are only comparable, it’s hard to tell whether the priors are really helpful or not.

6. There is a lack of an illustrative figure to show the main architecture of the proposed method, so the reader can only know how the idea is working till the end of the paper.

7. There are too many equations or derivations in the main text. Although they are clear and I appreciate the details, too many derivations in the main text make the paper lose its main focus.

8. There are inconsistent baseline names in text and tables, for example, 3D Geometry-aware Deformable Gaussians is called DAG in main text but GA3D in the table.

**Questions:**

Apart from the questions listed in weaknesses, here are some more questions:

1. Why the baseline numbers in the table not the same as the published version? According to the published version of D3G and GA3D papers, the performance of this paper is actually on the same level. For example, on D-NeRF datasets, D3G has 40.43 PSNR on average, while two types of the model have 40.36 and 40.54 on average. (lego scene is deleted due to data discrepancy, details in next question)

2. The lego scene is known that data discrepancy exists in the scenarios presented in the training and test sets. This can be substantiated by examining the flip angles of the Lego shovels. So some works, like D3G, try to evaluate the model on the validation split because evaluating on the test split is meaningless. Which split is evaluated here?

3. GA3D has reported its performance on HyperNeRF dataset, so why this baseline is deleted from the HyperNeRF experiments?

---

> ### Author Response · Authors · 2024-11-20
>
> We thank the reviewer for a detailed review.  Before addressing the specific weaknesses and questions raised, we would like to clarify and respond to some points made in the reviewer’s summary.
>
>
> **S1**: “This paper aims to solve dynamic reconstruction task. It proposes a new loss called ReMatching loss, in order to inject some physical priors into the learned deformation field”
>
> **A:** We do not claim to aim at solving the dynamic reconstruction task. Instead, acknowledging the inherently ill-posed nature of this problem—which necessitates the integration of priors—we propose a novel method for incorporating such priors (see lines 26–33). The goal of the ReMatching framework is not to “inject physical priors”, but rather to suggest a novel scheme to leverage priors to improve generalization to unseen frames, without compromising solutions’ fidelity levels (see lines 34-35). This is done by suggesting an optimization objective mimicking the APM (see lines 60-63) .
>
> **S2**: “ The paper builds the model based on the deformable-3d-gaussian paper, and tests the performance for three of the five categories”
>
> **A:** The suggested framework is not formulated with respect to a specific instantiation of a time-dependent function (see lines 38-39, 100-107). This is also supported empirically by applying our framework on time-dependent image pixel values (see lines 39, 510-515). The first two categories of priors, while not directly matching any of synthetic scenes due to their simplicity, are effectively integrated  and tested through the combined classes.
>
> **S3**:  “showing comparable performance to baselines.”
>
> **A**:  Our approach has advantages in reducing unrealistic distortions, such as visible issues in the dynamic movements of fingers in the jumping jacks scene and structural artifacts in *moving parts* like the leg in the T-Rex scene, the knife in the lemon scene, and the slicer in the banana scene. These points are discussed in lines 415-417 and 463-466 of the main text.
> Notably, our improvements primarily enhance image areas involving motion, which can skew
> the comparison based on the extent of movement presented within each scene. Nevertheless, our approach ranks first in 11 out of 13 scenes across at least two metrics.
> To strengthen these claims, the latest revision now includes additional qualitative evidence in the supplementary material, featuring  novel-view video reconstruction results comparing our model with baseline approaches.
>
> **W1:** In synthetic experiments, type IV and type V priors are selected for each scene. They should be thoroughly tested respectively to show the applicability under different scenarios.
>
> **A:** We respectfully disagree with the value of such an evaluation. Priors should be thoughtfully designed and applied in alignment with the specific dynamics exhibited by the scene. For instance, in scenes *without* static components, using the class  $P_V$  is less suitable (e.g., the jumping jack scene). Similarly, the classes  $P_I$  and  $P_{II}$ , while not directly matching any of D-NeRF synthetic scenes due to their simplicity, are effectively integrated through the combined classes  $P_{IV}$  and  $P_{V}$.
>
> **W2:** In real-world experiments, it is not clear which type of priors are used. And also, all of the priors should be tested to show the abilities.
>
> **A:** Thank you for this suggestion. We added an evaluation for both of the relevant priors $P_{III}$ and $P_{IV}$, incorporated in table 2 in the latest revision.
>
> **W3:** The velocity branches in the model only serve as a constraint part, but not used directly. The paper should demonstrate whether the learned velocity is meaningful or not. For example, using the velocity to advect the scene.
>
> **A:** We respectfully disagree with the reviewer’s suggestion. Our framework does not explicitly model a velocity branch as proposed. Employing a velocity field from the prior class to directly advect the scene is inconsistent with the principles of the ReMatching framework. The ReMatching loss is designed to optimize the dynamic reconstruction model, which operates via direct single-step evaluations only (see lines 142–143), to align as closely as possible with the prior class without compromising reconstruction fidelity. Importantly, elements from the prior class may inherently conflict with reconstruction quality (see lines 54–62 and 149–182 in the text for further explanation).

---

> ### Author Response · Authors · 2024-11-20
>
> **W4:** The segmentation compared with SAM is not proper and confusing. Since SAM segments objects according to semantics but this paper segments according to motions, it’s not proper to punish SAM’s over-segmentation.
>
> **A:** Our approach is *not* intended for semantic segmentation, nor do we make any claims in this regard. The qualitative comparison with SAM was conducted to highlight the non-triviality of the scene decomposition task. This comparison shows that, within the context of dynamic reconstruction, leveraging a foundation model such as SAM to generate scene decompositions that align with the underlying scene geometry may *not* always be the most effective approach.  Furthermore, including this comparison also adds value due to the fact that utilizing SAM for 3D-related tasks has gained popularity in recent works, such as [1] and [2].
>
>
> [1]: Zhou, Yuchen, et al. "Point-SAM: Promptable 3D Segmentation Model for Point Clouds." arXiv preprint arXiv:2406.17741 (2024).
>
> [2]: Cen, Jiazhong, et al. "Segment Anything in 3D with Radiance Fields." arXiv preprint arXiv:2304.12308 (2023).
>
> **W5:** Another question is about the bouncing ball example shown in the paper. As I know this scene includes some moving shadow on the ground, so the motion should not segment the shadow part as the same as the static ground. How is the segmentation applied?
>
> **A:** We understand the reviewer concerns, but it seems that there might be some confusion. We claim that shadow effects are *not* part of the geometry, but rather arise as a consequence of the specific geometric configuration and lighting conditions. To illustrate this, consider the reanimation of the bouncing balls scene: shadows are not independent entities capable of arbitrary deformation. The ability of our model to accommodate shadow effects into the static part (e.g., the floor in the bouncing balls scene) represents a significant success for our approach, highlighting its capability to improve the disentanglement of geometry from appearance.
>
> **W6:** The performances are only comparable, it’s hard to tell whether the priors are really helpful or not.
>
> **A:** Our approach effectively reduces unrealistic distortions, such as issues in the dynamic movements of fingers in the jumping jacks scene and structural artifacts in moving parts like the leg in the T-Rex scene, the knife in the lemon scene, and the slicer in the banana scene. These points are discussed in lines  415-417 and 463-466 of the main text. It is worth noting that our improvements primarily enhance image areas involving motion, which can skew the results based on the extent of movement presented in an image within each scene.
> To strengthen these claims, the latest revision now includes additional qualitative evidence in the supplementary material. This material features  novel-view video reconstruction results that compare our model with baseline approaches, providing a clearer illustration of the advantages highlighted in the main text.
>
> **W7:** There is a lack of an illustrative figure to show the main architecture of the proposed method, so the reader can only know how the idea is working till the end of the paper.
>
> **A:** Thank you for this suggestion. We added such a visualization in the implementation details in the appendix section, included in the latest revision. We note that our framework contribution is not in proposing an architecture, in fact applicable to several types of model functions.
>
> **W8:** There are too many equations or derivations in the main text. Although they are clear and I appreciate the details, too many derivations in the main text make the paper lose its main focus.
>
> **A:** We revised the method section to make it as concise as possible, which led to some simplifications in the mathematical descriptions of the derivations. However, we could not completely eliminate any of them. We would be happy to consider more specific examples if available.
>
> **W9:** There are inconsistent baseline names in text and tables, for example, 3D Geometry-aware Deformable Gaussians is called DAG in main text but GA3D in the table.
>
> **A:** Thank you for taking time to spot this. We fixed these inconsistencies, incorporated into the latest revision.

---

> > ### Author Response · Authors · 2024-11-20
> >
> > **Q1:** Why the baseline numbers in the table not the same as the published version? According to the published version of D3G and GA3D papers, the performance of this paper is actually on the same level.
> >
> > **A:** For D3G, we utilized the official public code and re-ran the evaluation. It’s important to note that when applying the ReMatching loss, we followed the exact same training protocol as D3G (see line 378), with the only difference being the inclusion of the ReMatching loss. For GA3D, we obtained the source code from the authors and re-ran the evaluation to account for two key differences from the original GA3D evaluation protocol: i) background pixel colors, and ii) image resolution (see lines 432-433). We assure the reviewers that we will make our code publicly available to allow further validation of this evaluation.
> >
> > **Q2:** The lego scene is known that data discrepancy exists in the scenarios presented in the training and test sets. This can be substantiated by examining the flip angles of the Lego shovels. So some works, like D3G, try to evaluate the model on the validation split because evaluating on the test split is meaningless. Which split is evaluated here?
> >
> > **A:** We followed the published protocol for the majority of our baselines, evaluating on the test set.
> >
> > **Q3:** GA3D has reported its performance on HyperNeRF dataset, so why this baseline is deleted from the HyperNeRF experiments?
> >
> > **A:** Thank you for the suggestion. We are currently in the process of evaluating GA3D using the source code provided by the authors on the HyperNeRF scenes considered in this work. We plan to include the results of this evaluation in the next revision, which will be uploaded before the end of the discussion period.

---

> ### Author Response · Authors · 2024-11-25
>
> We sincerely thank the reviewer once again for their detailed review.
>
> We have uploaded a second revision that includes the GA3D evaluation for the HyperNeRF dataset in Table 2. This additional result does not significantly alter the quantitative evaluation of this benchmark, with our approach still ranked first in at least two metrics across all five scenes.
>
> Furthermore, Section 8.4 now includes qualitative visualizations of the ReMatching framework’s ability to recover the underlying reconstruction flow. We hope this addresses the reviewer’s comment regarding the need for clearer visualizations. We are happy to provide further clarification if necessary.

---

> > ### Author Response · Authors · 2024-12-01
> >
> > We sincerely appreciate the time and effort you have dedicated to reviewing our work.
> >
> > We hope you had the opportunity to review our responses from November 19 and November 25. In these responses, we clarified the paper’s technical contributions and claimed benefits, provided a revision addressing evaluation concerns raised by the reviewer, included an illustrative figure of the architecture, and strengthened the qualitative evaluation by providing visualizations of novel-view video reconstructions in the supplementary materials, along with reconstruction flow/prior velocity field visualizations in the appendix that demonstrate our method's ability to recover the underlying flow.
> >
> > If there are any remaining questions or concerns, please let us know. We would be happy to provide further clarification.
> >
> > Thank you,
> >
> > The Authors

---

> > > ### Comment · Reviewer_GNax · 2024-12-02
> > >
> > > New quantitative and qualitative results show the effectiveness of this framework and address most of my concerns. However, I still have some suggestions for this paper.
> > >
> > > Generally speaking, more direct comparisons could be reorganized in the next version to improve the general complication and to make it more straightforward to understand.  Since the network backbone is D3G, it’s better to pose the results of D3G and your results tightly together, and highlight what is brought by your framework and your differences with D3G, cause this is the most direct baseline compared to others. It’s even better if you can add your pipeline to other Gaussian based pipelines like GA3D to show the similar performance gains.
> > >
> > > For W1, what I want to see is a thorough presentation of the model’s performance either when a proper or not proper prior is assigned, because a scene may include various priors in many real-world cases. So I still insist on my suggestion in W1, you don’t need to be SOTA for all the cases, but you should show the ability of your method.
> > >
> > > For W3, since you have a set of learned weights and a predefined basis, if the quantitative comparison is not easy, it’s practical to at least qualitatively visualize the velocity rematched, for example, to visualize it as scene flow, or some other better ways. This can greatly help readers understand what is the underlying prior adapted by the framework. This is what you are trying to do in Figure 1, but should be improved.
> > >
> > > For W4, I agree that comparing the performance with SAM adds value, but I mean it’s not fair to punish the over-segmentation of SAM. For example, it at least shows the geometry for hands and other objects but your model cannot. So it’s worth giving a more thorough analysis of what kind of motion can be segmented with priors but what kind of motion cannot, which is clear in your response to my W5.
> > >
> > > For W6, for my original words “main architecture”, what I mean is an illustration of the overall pipeline.
> > >
> > > Considering that most of my concerns have been addressed, I would increase my rating to 5.
> > >
> > > Since the time left is limited for you due to my late reply, please briefly respond to my above comments with your commitment to the next version. I'll finalize my rating after receiving your feedback if available.

---

> > > > ### Author Response · Authors · 2024-12-02
> > > >
> > > > We thank the reviewer for their effort in reviewing our response. We are pleased that most of the reviewer’s concerns have been addressed, and we appreciate their recognition of the effectiveness of the proposed framework.
> > > >
> > > > We also thank the reviewer for their additional insightful suggestions. Below is our response:
> > > >
> > > > Thank you for clarifying your **general concern**. Based on the reviewer's suggestion, we will revise Tables 1 and 2 in the next version to improve clarity. Specifically, we will demonstrate the application of our framework on an **alternative dynamic reconstruction** pipeline, such as GA3D, and rename the evaluation to emphasize this characteristic of our framework. For your review, here is a link to the proposed table structure: https://imgur.com/a/HQcuroB.
> > > >
> > > > In response to the reviewer’s comment, we would also like to reiterate that the paper already demonstrates the framework’s ability to effectively control **other types** of model functions, such as time-dependent image intensity values. This highlights that the applicability of the framework is relatively independent of the dynamic scene representation, making it potentially beneficial for a wide range of models.
> > > >
> > > > Thank you for clarifying **W1**. In the **next** revision, we will enhance the ablation study in Section 8.6. This study already evaluates the simple prior class $P_2$ by addressing the case $k=1$ within $P_4$, demonstrating the expected improvement of the adaptive prior class $P_4$ when $k$ is set larger than the anticipated number of moving parts. We will **extend** this analysis to incorporate the simple prior class $P_1$​ as well.
> > > >
> > > > Thank you for clarifying **W3**. The paper includes visualizations addressing the review’s concerns in the following:
> > > > - Figure 10: Comparison of the learned ReMatched velocity field against the (unknown) ground truth.
> > > > - Figure 13: Comparison of the learned ReMatched velocity field against the Matched velocity field derived from the prior class.
> > > > - Supplementary material (2:27–2:48 in the video): Colored geometry following the learned time-dependent function, clearly demonstrating that our method effectively recovers the underlying reconstruction flow of the shovel movement in the lego scene.
> > > >
> > > > **In the next revision**, we will clarify the text to better reference these visualizations and explain their role in illustrating the rematched velocity fields.
> > > >
> > > >
> > > > Thank you for clarifying **W4**. In the **next revision**, we will improve the text to clarify this property of our framework: it is capable of recovering scene decompositions based on the underlying movements observed in the scene images. Contrary to the reviewer’s suggestion, SAM does **not only** over-segment scenes; in some instances, it also merges parts that should move **independently** into a **single** part. For example, in the “jumping jacks” scene, the arms are merged with the body because the subject is wearing a sweater of the same color: https://imgur.com/a/zVWUmdk. We will include this result and clarify this analysis in the **next revision** as well.
> > > >
> > > > Thank you for clarifying **W6**. In the **next revision**, we will enhance Figure 5 by adding small arrows to illustrate the interface with the ReMatching loss.
> > > >
> > > > If there are any remaining questions or concerns, please let us know. We would be happy to provide further clarification.
> > > >
> > > > Thank you,
> > > >
> > > > The Authors

---

### Official Review · Reviewer_zqNk · 2024-11-04

**Soundness:** 3
**Presentation:** 3
**Contribution:** 2
**Rating:** 6
**Confidence:** 3

**Summary:**

The paper introduces the ReMatching framework that provides deformation priors into dynamic reconstruction models. It can be adopted into multiple types or model priors and improve dynamic reconstruction accuracy over previous methods.

---

I rate the paper marginally below the acceptance threshold. The paper provides theoretical backgrounds and shows that it works, but it's not so clear what specific problem the paper is trying to tackle and overcome what challenges in the literature.

**Strengths:**

* Theoretical background

   The method provides theoretical background on its proposed representation. The supplemental further provides proof of lemma for better understanding.

* Good accuracy

  The method shows better accuracy than other previous works in general (depending on the evaluation datasets and metrics though).

**Weaknesses:**

* Killing application?

   The method theoretically sounds good, but it seems not so clear what problem the method is mainly targeting. Are there any problems or failing cases that the proposed method solves whereas the previous method fails? Experiments (Table 1 and 2, Figure 2) show the benefit of the method for better rendering quality, but are there any major challenges that the method targets to solve?

* Improvement in generalization quality?

   The paper claims about the improvement in generalization quality, but the experiment shows that it doesn't clearly outperform previous methods, but is competitive or slightly better. In what sense does the paper improve generalization quality? I hope it's more clearly explained.

**Questions:**

* Hyperparameter choice

   How is the method sensitive to the hyperparameter choice for the main objective loss and the entropy loss?

* (minor)

   Sometimes equations are referred to by just their numbers. In line 224, "problem from 5 with 9" $\rightarrow$ "problem from equation (5) to equation (9)". It would be good to double-check and revise them between Page 3 and Page 7.

---

> ### Author Response · Authors · 2024-11-20
> **Response to Reviewer Comments**
>
> We thank the reviewer for a detailed review. Below we address the main comments and questions expressed in this review.
>
> **W1:** It's not so clear what specific problem the paper is trying to tackle and overcome what challenges in the literature.
>
> **A:** We divide our answer into two parts.
>
> “what specific problem the paper is trying to tackle”
>
> The paper aims to improve the accuracy of dynamic reconstruction models, a long-standing open research challenge. This difficulty is exemplified in the qualitative results of existing baselines showcased in the paper, which highlight their limitations in capturing realistic dynamics. The core challenge arises from the ill-posed nature of the task: dynamic reconstruction relies on discrete (sparse) samples of a continuous phenomenon (e.g., objects moving continuously over time). This inherent sparsity makes it essential to incorporate prior knowledge to guide any reconstruction model. This answer can be found in the text in lines 26-34.
>
> “overcome what challenges in the literature”
>
> The Ill-posedness of the reconstruction problem necessitates the use of prior knowledge, which is inherent to any algorithm addressing this problem. However, since in many cases priors do not match exactly, enforcing prior assumptions for many existing methods often compromises fidelity to the input data. Our work tackles this issue not by proposing new types of priors—already abundant in the literature—but by introducing the ReMatching loss, which optimizes for solutions that remain as close as possible to the desired prior class, rather than strictly enforcing membership in the prior class. This approach minimizes the tradeoff, preserving fidelity while benefiting from the priors.
>
> Additionally, we address two more challenges:
>
> Generality across models: Our solution is designed to be independent of any specific reconstruction model. This generality allows the framework to unify diverse components, such as geometry representations and image rendering functions, under a common approach. For instance, by extending its applicability to rendered images, our framework can be incorporated to the training of any differentiable dynamic reconstruction model, making future advancements in this framework broadly relevant to many models.
>
> Flexibility in prior design: We enable users of the framework to mix prior classes to tailor the method to their specific instance of the reconstruction problem, enhancing its adaptability across different scenarios.
>
> This answer can be found in the text in lines 34-42..
>
> **W2:** “Killing application? The method theoretically sounds good, but it seems not so clear what problem the method is mainly targeting… Experiments (Table 1 and 2, Figure 2) show the benefit of the method for better rendering quality, but are there any major challenges that the method targets to solve?”
>
> **A:** Models enabling photorealistic renderings of novel views are pivotal in fields like the film industry, video gaming, and AR/VR. Their utility extends beyond entertainment also to scientific fields like biology, physics, and geography, where accurate visualization of dynamic phenomena is crucial (see [1] for further discussion).
> Moreover, dynamic reconstruction models play an essential role in *downstream* applications such as scene editing and scene reanimation. For example, reanimation is possible by utilizing the model’s learned geometry to control novel, plausible movements. Improving dynamic reconstruction models, as our work aims to do, has the potential to significantly benefit such downstream applications as well. Our evaluation on novel-view rendering provides a robust proxy for assessing the quality a model achieves in learning meaningful geometry and appearance representations of dynamic scenes. Thus, the improvements in novel-view rendering of our approach hold promise for improving the quality of downstream applications as well—which we identify as an important future research direction.
>
> [1]: Yunus, Raza, et al. "Recent Trends in 3D Reconstruction of General Non‐Rigid Scenes." Computer Graphics Forum. 2024.

---

> > ### Author Response · Authors · 2024-11-20
> > **Response to Reviewer Commnts**
> >
> > **W3:** “Improvement in generalization quality?... In what sense does the paper improve generalization quality? I hope it's more clearly explained”.
> >
> > **A:** In this work, generalization of a dynamic reconstruction model refers to the model’s ability to generate accurate results for novel-view images and timestamp frames that were not included in the training data. The evaluations presented in this paper on frames unseen during training demonstrate that our approach effectively reduces unrealistic distortions, such as issues in the dynamic movements of fingers in the jumping jacks scene and structural artifacts in moving parts like the leg in the T-Rex scene, the knife in the lemon scene, and the slicer in the banana scene. These points are discussed in lines 441-443 and 495-497 of the main text.
> > To strengthen these claims, the latest revision now includes additional qualitative evidence in the supplementary material. This material features  novel-view video reconstruction results that compare our model with baseline approaches, providing a clearer illustration of the advantages highlighted in the main text.
> >
> > **Q1:** Hyperparameter choice.How is the method sensitive to the hyperparameter choice for the main objective loss and the entropy loss?
> >
> > **A:** Thank you for this question, we agree with the reviewer on the importance of testing a range of values for the ReMatching loss weight $\lambda$ and the entropy loss. To address this, we included such an analysis in the appendix of the latest revision. The results indicate that:
> >
> > i) The loss weight $\lambda$ is robust within the range $[5\mathrm{e}{-4} , 5\mathrm{e}{-3}]$, consistently improving results, while $\lambda \lt 1\mathrm{e}{-5}$ aligns with the baseline. Larger values, $\lambda \gt 1\mathrm{e}{-2}$, may compete with the reconstruction loss, leading to suboptimal solutions.
> >
> > ii) The entropy loss provides a robust improvement over the baseline in the majority of the metrics for a set of weights within the range $[1\mathrm{e}{-4} , 1\mathrm{e}{-3}]$.
> >
> > **Q2:** Sometimes equations are referred to by just their numbers. It would be good to double-check and revise them between Page 3 and Page 7.
> >
> > **A:** Thank you for taking the time to spot this. We have corrected these inconsistencies in the latest revision.

---

> > > ### Comment · Reviewer_zqNk · 2024-11-28
> > >
> > > Thanks for the response. I updated my rating to marginally above the acceptance threshold. The response helped me better understand the underlying motivation and clarity of the proposed method.

---

> ### Author Response · Authors · 2024-12-01
>
> Thank you for reviewing our rebuttal and updating the score!
>
> Thank you,
>
> The Authors

---

### Official Review · Reviewer_5yAB · 2024-11-05

**Soundness:** 3
**Presentation:** 3
**Contribution:** 3
**Rating:** 6
**Confidence:** 3

**Summary:**

This paper tackles the challenge of dynamic scene reconstruction by introducing a novel loss function to incorporate diverse deformation priors into a unified framework. This new loss uses velocity-field-based priors and solves as a flow-matching problem. This design allows the framework to integrate a variety of deformation priors, such as piece-wise rigid transformations and volume-preserving deformations, enhancing the flexibility and generalization capabilities of existing models such as Gaussian Splatting. In experiments, the authors validate the robustness of their method across multiple benchmarks, demonstrating consistent improvements over baselines in both synthetic and real-world dynamic scenes.

**Strengths:**

- The paper introduces the ReMatching loss, designed to enhance dynamic scene reconstruction by leveraging deformation priors in a unified, adaptable way. This approach directly addresses limitations in generalization for dynamic NVS.
-  ReMatching focuses on incorporating velocity-field-based priors to better model deformations over time, enabling seamless integration with existing dynamic reconstruction models. The framework supports various types of deformation priors, including restricted deformations, piece-wise rigid transformations, and volume-preserving deformations.
- A notable feature of this framework is its capacity to combine multiple priors, allowing for the construction of more complex deformation classes. This adaptability is key for addressing the diverse requirements of real-world dynamic scenes.
- By using the Gaussian Splatting image model as a backbone, the framework consistently outperforms existing methods across multiple benchmarks, demonstrating improved reconstruction accuracy and robustness in both synthetic and real-world dynamic scenes. It shows that ReMatching is not only theoretically sound but also empirically robust across various challenging scenarios.

**Weaknesses:**

- The proposed framework and loss function include some details and design choices, many of which seem crucial for achieving optimal performance. However, no ablation study is presented to isolate the effects of these parameters or design decisions, leaving it unclear how each contributes to the final results.
- Introducing a new loss function raises questions about its impact on training stability and convergence speed, but these aspects are not explored in the paper. Additional insights into how this loss affects training dynamics would provide valuable context for readers, such as  runtime comparisons, convergence plots, or error analysis across iterations.
- The paper asserts that the method is simulation-free, meaning that each evaluation of $\psi_t$ requires only a single step, avoiding potential numerical instabilities. However, it's unclear if this claim has been experimentally validated or remains a theoretical assumption. If any simulation was conducted, sharing empirical evidence comparing the stability and computational efficiency would clarify the stability and feasibility of this approach.
- From the quantitative results, the improvements over the second-best method are relatively modest. It remains unclear what the primary advantage of this approach is—perhaps robustness? Clarifying this would help readers understand the main strengths of the proposed method.
- Additionally, there are numerous dynamic NeRF datasets and benchmarks beyond the synthetic, 360-degree dynamic NeRF dataset, such as those used in NSFF and DynIBaR, which feature more complex, real-world, forward-facing scenes. It would be insightful to see how well the proposed loss function performs on these more challenging datasets, as it would demonstrate its applicability to diverse, real-world scenarios.

**Questions:**

- Would it be possible to apply the new loss function directly to a baseline method without altering any other components? This would allow for a strong, fair comparison that isolates the effect of the proposed loss. For example, the authors can try conducting an ablation study where ReMatching loss is applied to a baseline method (e.g., D3G) without any other modifications, and report the results. This would help isolate the impact of the proposed loss function.
- L215 "we assume that P are linear sub-spaces, hence allowing for fast solvers with a runtime of at most O(n)". Could the authors clarify why it is reasonable to assume P is linear and specify the fast solvers used? This additional detail would help readers understand the computational benefits and assumptions behind this approach.

---

> ### Author Response · Authors · 2024-11-20
> **Response to Reviewer Comments**
>
> We thank the reviewer for a detailed review. Below we address the main comments and questions expressed in this review.
>
> **W1:** no ablation study is presented to isolate the effects of the loss function parameters.
>
> **A:** Thank you for this suggestion. We note that some of the effects of the ReMatching loss are evaluated in the paper through a comparison with D3G (see Tables 1 and 2), as D3G uses the same rendering model and training protocol, making the ReMatching loss the only difference. Additionally, the fact that we used a consistent hyperparameter value of $\lambda = 0.001$ (see line 386) across all experiments suggests a positive indicator of the ReMatching loss’s robustness.
>
> However, we agree with the reviewer on the importance of a more comprehensive ablation study, testing a range of values for the ReMatching loss weight $\lambda$ and, in cases involving the adaptive prior class as in $\mathcal{P}_{\mathrm{IV}}$, the maximum number of possible pieces $k$. To address this, we included such an analysis in the appendix of the latest revision. The results indicate that:
>
> i) The loss weight $\lambda$ is robust within the range $[5\mathrm{e}{-4} , 5\mathrm{e}{-3}]$, consistently improving results, while $\lambda \lt 5\mathrm{e}{-5}$ aligns with the baseline. Larger values, $\lambda \gt 1\mathrm{e}{-2}$, may compete with the reconstruction loss, leading to suboptimal solutions.
>
> ii) The ablation study for the parameter  $k$  demonstrated stable performance across a relatively wide range of choices, offering flexibility in selecting  $k$ based on leveraging prior knowledge about the expected number of moving parts in the scene. This, in our view, makes $k$ a relatively user-friendly hyperparameter to configure for effective use of the loss.
>
> **W2:** A new loss function raises questions about its impact on training stability and convergence speed. Additional insights such as runtime comparisons, convergence plots, or error analysis across iterations.
>
> **A:** Thank you for this suggestion. We agree with the reviewer on the value of such analysis, which we have now included in the appendix of the latest revision.  Regarding runtime comparisons, the results validate our claims about the additional computational cost (see lines 214-215) compared to the baseline. For the convergence analysis, the results indicate that the ReMatching loss exhibits similar qualitative behaviour to the baseline reconstruction loss.
>
> **W3:** The paper asserts that the fact that the method is simulation-free helps avoid potential numerical instabilities. However, it's unclear if this claim has been experimentally validated or remains a theoretical assumption.
>
> **A:** We agree with the reviewer that our previous statement on the limitations of simulation needed refinement. Indeed, a more accurate characterization is that the error in flow simulation correlates with the computational budget allocated to the simulation steps (i.e., in the idealized case of an infinite computational budget, simulation error would be eliminated). Thus, a more precise claim is that our approach, being simulation-free, fully avoids the need for potentially costly simulation procedures. This clarification has been incorporated into the latest revision.

---

> > ### Author Response · Authors · 2024-11-20
> > **Response to Reviewer Comments**
> >
> > **W4:** From the quantitative results, the improvements over the second-best method are relatively modest. It remains unclear what the primary advantage of this approach is.
> >
> > **A:**  We note that finding reliable quantitative metrics for novel-view rendering remains an open research problem. For instance, PSNR focuses on pixel-level errors, which can lead to high scores even when unnatural textures or structural artifacts—such as blurring in dynamic scenes—are present [1]. For further discussions on metrics limitations, see also references [2] and [3].
> >
> > Moreover, our improvements primarily enhance image areas involving motion, which can skew
> > the quantitative results based on the extent of movement presented in an image within each scene. Despite these challenges, our approach ranks first in 11 out of 13 scenes across at least two metrics. For the Lego scene, we discovered that the K-Planes evaluation was conducted at a lower resolution (400x400) compared to the 800x800 resolution used here and in other baselines. Upon re-evaluating K-Planes at 800x800, we obtained the results: 0.0472 (LPIPS), 25.15 (PSNR), 0.9431 (SSIM),  further solidifying our approach as the best in 12 out of 13 scenes. This update will be included in the next revision.
> >
> > We note that qualitatively, our approach has practical advantages in reducing unrealistic distortions, such as visible issues in the dynamic movements of fingers in the jumping jacks scene and structural artifacts in moving parts like the leg in the T-Rex scene, the knife in the lemon scene, and the slicer in the banana scene. These points are discussed in lines 415-417 and 463-466 of the main text. To strengthen these claims, the latest revision now includes additional qualitative evidence in the supplementary material, featuring  novel-view video reconstruction results comparing our model with baseline approaches.
> >
> > [1] : Wang, Zhou, et al. "Image quality assessment: from error visibility to structural similarity." IEEE transactions on image processing 13.4 (2004): 600-612.
> >
> > [2]: Wang, Zhou, and Alan C. Bovik. "Mean squared error: Love it or leave it? A new look at signal fidelity measures." IEEE signal processing magazine 26.1 (2009): 98-117.
> >
> > [3]: Sjögren, Oskar, et al. "Identifying and Mitigating Flaws of Deep Perceptual Similarity Metrics." arXiv preprint arXiv:2207.02512 (2022).
> >
> > **W5:** There are numerous dynamic NeRF datasets and benchmarks beyond the synthetic, 360-degree dynamic NeRF dataset, such as those used in NSFF and DynIBaR, which feature more complex, real-world, forward-facing scenes.
> >
> > **A:**  Thank you for this suggestion. In response, we selected four scenes from the dynamic scenes dataset  suggested by the reviewer, and performed both quantitative and qualitative comparisons to the D3G baseline. We note that none of the baselines considered in this work have conducted evaluations on this dataset. We are pleased to report that our results demonstrate consistent improvements over D3G, similar to those observed on both the synthetic D-Nerf and real-world HyperNeRF datasets. This additional analysis is included in the appendix of the latest revision.
> >
> > **Q1:** Would it be possible to apply the new loss function directly to a baseline method without altering any other components?
> >
> > **A:**  This approach was applied as outlined in the paper (see lines 373 and 378). We confirm for the reviewer that the only implementation difference between our method and D3G is the incorporation of the ReMatching loss into the optimization objective.
> >
> > **Q2:** Could the authors clarify why it is reasonable to assume P is linear and specify the fast solvers used?
> >
> > **A:** In our work, we can parameterize the prior class as a linear subspace because linear parameterizations are sufficiently expressive to encompass the entire class for the type of priors considered. Unlike methods requiring more complex models, such as neural networks, our approach benefits from defining prior classes via velocity fields, enabling straightforward linear characterizations. For instance,  $div(u_1) = 0$ and $div(u_2) = 0$ implies that $div(u_1 + u_2) = 0$ (see line 270), making a Fourier-like linear basis, as used in  $\mathcal{P}_{III}$ , expressive enough.
> > Similarly, since any rigid motion is generated by a velocity field of the form $v(x) = Ax + b, A^T = A$, it suffices to consider the linear space defined by  $A = A^T$  (see lines 244-247).
> >
> > For all cases in this paper, solving equation (5) reduces to solving a linear system of equations, which can be efficiently managed using standard operations in the PyTorch library (specifically, by inverting a $d \times d$ matrix). We agree with the reviewer that providing these details enhances the clarity of our work and have added them to the latest revision (see lines 212-215).

---

> > > ### Author Response · Authors · 2024-12-01
> > >
> > > We sincerely appreciate the time and effort you have taken to review our work.
> > >
> > > We hope you had the opportunity to review our response, where we provided additional evaluations on the Dynamic Scene datasets recommended by the reviewer. These evaluations further highlight the applicability of our method to real-world scenarios and offer additional evidence supporting the robustness of the framework’s parameters. Additionally, we included an ablation study on key hyperparameters for further insights.
> > >
> > > We are happy to provide further clarification on any remaining questions or concerns.
> > >
> > > Thank you,
> > >
> > > The Authors

---

### Author Response · Authors · 2024-11-20

We sincerely thank the reviewers for their time and thoughtful feedback. In response to their suggestions, we have uploaded a revised version of the paper, with changes highlighted in blue. The key updates include:

**Additional Evaluation:** A new evaluation on a real-world dataset is included in the appendix.

**Ablation Study:** A study of key hyperparameters in the ReMatching framework is included in the appendix.

**Qualitative Comparisons:** To better illustrate the advantages highlighted in the main text, supplementary material now includes novel-view video reconstruction results comparing our model with baseline methods.

**Derivations:** Details explaining the transition from equation (4) to equation (9) are included in the appendix.

Additionally, we plan to upload another revision before the end of the discussion period, incorporating the visualization requested by reviewer SSgx. Finally, to address some of the reviewers’ concerns regarding evaluation, we assure the reviewers that we are committed to publicly releasing the experiment's code upon acceptance.

Detailed responses addressing each review are provided individually.

---

### Author Response · Authors · 2024-11-25

We would like to once again thank the reviewers for their valuable and constructive feedback.

In response to the insightful suggestions and concerns raised during the review process, we have uploaded a **second** revision that primarily addresses issues related to visualizing the framework’s claimed properties and improving the clarity of the contributions.

Below, we have summarized the key rebuttal points raised by reviewers during the discussion period and our revisions relating to each point:

**Missing visualization of learned reconstruction flows.** The revised paper now includes a qualitative evaluation in Section 8.4, showcasing the ReMatching framework’s ability to align with the underlying reconstruction flow. This section also highlights the importance of the reprojection procedure, demonstrating that although not all velocity fields in the prior class are suitable for guiding the optimization, the ReMatching loss ensures convergence to an appropriate prior. This enables an accurate recovery of the reconstruction flow.

**Missing a clear statement of contributions.** The revised paper now includes a new paragraph in the introduction to clarify the contributions, making it easier for the reader to quickly discern the main strengths of our approach.

**Qualitative Comparisons.** To better illustrate the advantages highlighted in the main text, supplementary material includes novel-view video reconstruction results comparing our model with baseline methods. These provide a clear demonstration of the effects of the ReMatching loss on the spatial and temporal consistency of the scene reconstruction.

**Robustness of the framework hyperparameters.** The framework’s hyperparameters were set to fixed values across all experiments in this work. To further address concerns, we have added an evaluation on a new real-world dataset, the Dynamic Scenes dataset, where the same hyperparameters produced improvements consistent with the other benchmarks. Additionally, we conducted an ablation study on key hyperparameters, demonstrating expected behavior. These results are included in the appendix of the revised paper.

**Computational efficiency.** The revised paper includes timing evaluations to support our claims about the computational efficiency of the loss function.

**Additional Evaluation.** We have introduced a new evaluation using the Dynamic Scenes dataset (Yoon et al., 2020), which features forward-facing views of real-world scenes with complex dynamics. The results highlight improvements in fine details, such as the truck’s front lights and the bottom teeth of dynamic faces. Furthermore, the approach minimizes reconstruction motion artifacts, as observed in the motion of jumping humans and the dinosaur’s head and legs in the balloon scene. Detailed results are presented in Section 8.3 of the appendix.

**Evaluation details.** Concerns regarding the evaluation setup (e.g., missing baselines and testing protocols) have been addressed in the revised paper. We also reaffirm our commitment to making the experimental code publicly available to support reproducibility.

Overall, the review process has been encouraging, with reviewers recognizing the importance of the problem we tackle (SSgx, 5yAB) and appreciating the technical contribution of the proposed approach (zqNk, SSgx, 5yAB). The originality of our framework in addressing dynamic novel-view synthesis as a prior-guided flow-matching problem has been acknowledged (SSgx), and reviewers have validated the improvements in the reconstruction accuracy the method provides (5yAB, zqNk). We deeply appreciate the thoughtful reviews and believe that the revisions have resulted in a stronger paper.

With the remaining time in the discussion period, we would welcome any additional feedback from the reviewers and are happy to address any remaining concerns.

---

### Meta-Review · Area_Chair_Lf6f · 2024-12-21

**Metareview:**

This paper proposes ReMatching, a novel framework for dynamic scene reconstruction that leverages velocity-field-based priors to enhance generalization quality. Reconstructing dynamic scenes from image inputs is an important task in computer vision with numerous practical applications, yet existing methods often fall short in achieving high-quality reconstructions from unseen viewpoints and timestamps. The proposed method effectively addresses these challenges by introducing deformation priors and a matching procedure that can seamlessly integrate with existing dynamic reconstruction pipelines.

All reviewers provided positive ratings, with one reviewer (Reviewer GNax) indicating a rating increase during the discussion phase. This reflects the consensus regarding the quality and impact of the paper. After reviewing the rebuttal and discussions, the area chair agrees with the reviewers’ assessments and finds no reason to overturn the consensus, recommending the paper for acceptance.

Several reviewers suggested that the paper could benefit from explicitly stating its core contributions in a way that is easily accessible to readers. Doing so would help to better communicate the significance of the work. While the paper is technically sound, reviewers encouraged the authors to refine the writing in the final version to ensure clarity and readability, particularly for readers less familiar with the nuances of dynamic scene reconstruction. The authors are also encouraged to incorporate other suggested improvements in the final version to maximize the paper’s impact.

**Additional Comments On Reviewer Discussion:**

The authors’ rebuttal addressed the reviewers’ concerns by providing more detailed explanations of the contributions of the work. As a result, Reviewers GNax and zqNk have increased their ratings to borderline accept. In response to the concern that the performance improvement is marginal, the rebuttal presented additional quantitative and qualitative results, which help strengthen the submission.

---

### Decision · Program_Chairs · 2025-01-22

Accept (Poster)